# Text-to-Image Rectified Flow as Plug-and-Play Priors

**Xiaofeng Yang[1], Cheng Chen[1], Xulei Yang[2], Fayao Liu[2], Guosheng Lin[1]**[*]
[1]College of Computing and Data Science, Nanyang Technological University, Singapore
[2]Institute for Infocomm Research, A*STAR, Singapore
`yang.xiaofeng@ntu.edu.sg, gslin@ntu.edu.sg`

## Abstract

Large-scale diffusion models have achieved remarkable performance in generative tasks. Beyond their initial training applications, these models have proven their ability to function as versatile plug-and-play priors. For instance, 2D diffusion models can serve as loss functions to optimize 3D implicit models. Rectified Flow, a novel class of generative models, has demonstrated superior performance across various domains. Compared to diffusion-based methods, rectified flow approaches surpass them in terms of generation quality and efficiency. In this work, we present theoretical and experimental evidence demonstrating that rectified flow based methods offer similar functionalities to diffusion models — they can also serve as effective priors. Besides the generative capabilities of diffusion priors, motivated by the unique time-symmetry properties of rectified flow models, a variant of our method can additionally perform image inversion. Experimentally, our rectified flow based priors outperform their diffusion counterparts — the SDS and VSD losses — in text-to-3D generation. Our method also displays competitive performance in image inversion and editing. Code is available at: https://github.com/yangxiaofeng/rectified_flow_prior.

## 1 Introduction

Recent advances in diffusion models (Ho et al., 2020; Song & Ermon, 2019; Rombach et al., 2022) have revolutionized generative tasks in image, video, and music production (Dhariwal & Nichol, 2021; Guo et al., 2023b; Huang et al., 2023), often surpassing GANs (Goodfellow et al., 2014; Karras et al., 2019). Beyond excelling in tasks for which they are specifically trained, various studies (Graikos et al., 2022; Poole et al., 2022; Wang et al., 2023; Yu et al., 2024) indicate that these models can also serve as plug-and-play priors for related tasks, i.e., using the pretrained diffusion models as loss functions. For instance, Dreamfusion (Poole et al., 2022) introduces the SDS loss, utilizing a pretrained text-to-image diffusion model to generate 3D objects. Several subsequent works (Hertz et al., 2023; Wang et al., 2023; Yu et al., 2024; Katzir et al., 2024; Yang et al., 2023) have refined the SDS loss to further enhance generation quality and diversity, applying diffusion priors to a variety of downstream applications, including 2D/3D editing and 4D generation.

Rectified flow methods (Liu et al., 2022; Albergo & Vanden-Eijnden, 2022), which connect data and noise via straight-line trajectories, are emerging as promising alternatives to diffusion models. Early works like InstalFlow (Liu et al., 2024) demonstrate high-quality image synthesis in few steps. Recent advancements (Ma et al., 2024; Esser et al., 2024) achieve state-of-the-art results in text-to-image generation. Given this growing research focus, it remains to be investigated whether large-scale pretrained rectified flow models can function as effective priors, similar to diffusion-based methods.

In this work, we explore how to distill knowledge from pretrained text-to-image rectified flow models and use them as priors for various tasks. To ensure a broad applicability, our approach is based on the generalized Stochastic Interpolants framework (Albergo et al., 2023), rather than being directly tied to the rectified flow formulation. By incorporating specific interpolation factors of rectified flow, we adapt our method to this model type.

---

[*]Corresponding Author

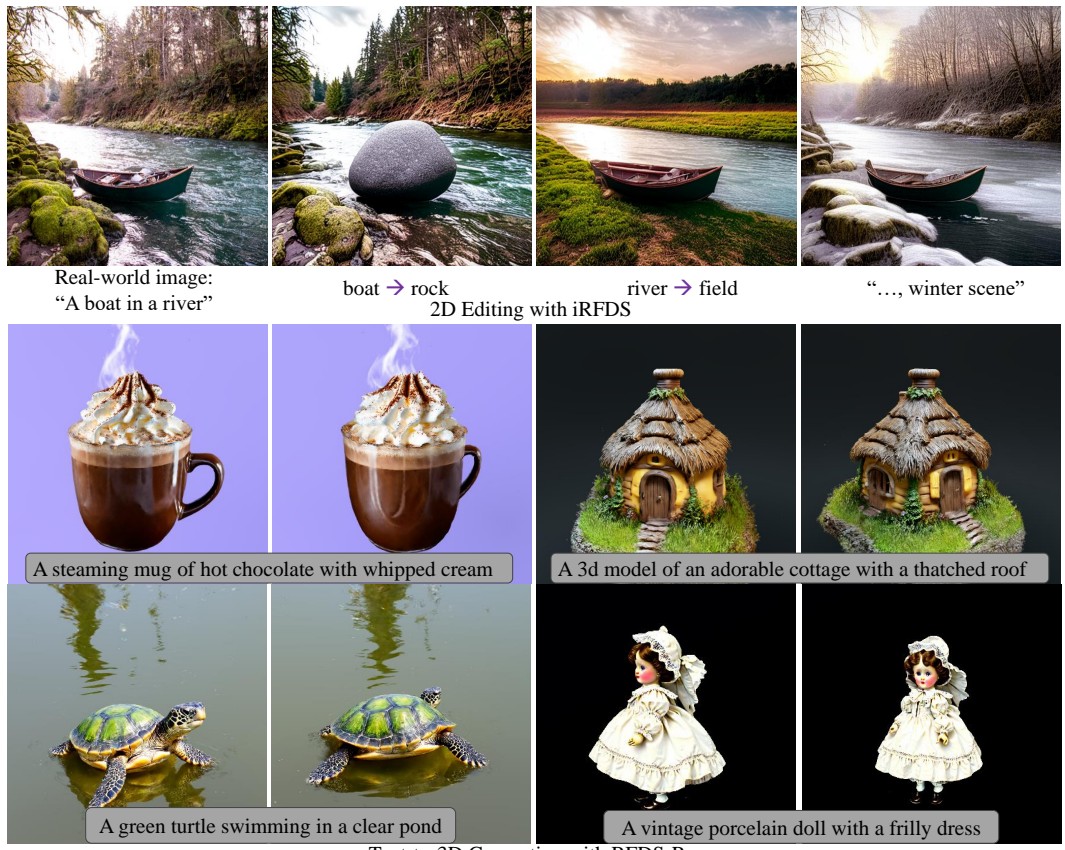

Real-world image:
"A boat in a river"    boat → rock    river → field    "…, winter scene"

2D Editing with iRFDS

A steaming mug of hot chocolate with whipped cream    A 3d model of an adorable cottage with a thatched roof

A green turtle swimming in a clear pond    A vintage porcelain doll with a frilly dress

Text-to-3D Generation with RFDS-Rev

Figure 1: Demonstration of potential use cases of the proposed rectified flow priors. Our methods can be used on 2D inversion and editing and text-to-3D generation.

We begin by proposing RFDS (**R**ectified **F**low **D**istillation **S**ampling), a baseline method analogous to the SDS loss in diffusion models. We derive the baseline RFDS by calculating the gradients of the input image by reversing the flow-matching training process. We observe a phenomenon similar to that seen in diffusion-based methods: by removing the Jacobian of the rectified flow network, RFDS can generate meaningful images or 3D objects given text conditions.

Moreover, inspired by the straight line trajectories of rectified flow, we notice that the baseline RFDS can be extended to optimize input noise by taking the "negative gradient" of the original RFDS. This extension, referred to as iRFDS (inverse RFDS), is particularly valuable for tasks such as image inversion and editing, where real images are mapped back to their latent noise representations before being edited with modified text prompts. Experiments show that iRFDS achieves high-quality results in image editing and inversion.

Finally, we propose RFDS-Rev (RFDS Reversal) to overcome the limitations of the baseline RFDS. Similar to the SDS loss, the original RFDS loss tends to produce outputs with limited details and diversity. We analyze this behavior through both intuitive reasoning and mathematical explanation. To address this issue, RFDS-Rev employs an iterative process that alternates between using iRFDS to recover the original noise and applying RFDS to refine the input. This dual-step process significantly improves the quality of the generated content, enabling RFDS-Rev to outperform the baseline RFDS method and establish new benchmarks in downstream applications.

Additionally, we demonstrate that our method is applicable to score-matching methods, such as diffusion models. The diffusion models, trained with score-matching objectives, can be reformulated as deterministic PF-ODEs, making them compatible with our proposed methods. Notably, when we formulate score-matching models in terms of PF-ODEs, our proposed RFDS baseline becomes identical to the SDS loss. Furthermore, our introduced iRFDS and RFDS-Rev expand upon previ-

ous SDS-like, score-matching-based knowledge distillation approaches, broadening their scope and applicability.

Our experiments are conducted on 2D image inversion and editing for iRFDS and text-to-3D generation for RFDS and RFDS-Rev. Notably, the RFDS-Rev method achieves a new state-of-the-art performance in text-to-3D benchmarks among all 2D lifting methods, surpassing a broad spectrum of diffusion priors. Some generation results can be found in Fig. 1.

To summarize our contributions:

- We propose the first series of studies that utilize pretrained rectified flow models as priors. Our methods span a wide array of applications, from generative tasks such as text-to-3D generation to the inversion and editing of real images.

- Compared to diffusion-based priors, our methods significantly outperform them in text-to-3D generation and achieve highly competitive results in image inversion and editing.

## 2 PRELIMINARIES: FLOW-BASED MODELS AND RECTIFIED FLOW

The idea of rectified flow is proposed in multiple concurrent works (Liu et al., 2022; Lipman et al., 2022; Albergo & Vanden-Eijnden, 2022). Here, we follow the interpretation of Stochastic Interpolants (Albergo & Vanden-Eijnden, 2022; Ma et al., 2024) to briefly introduce the basic concepts.

The Stochastic Interpolants framework starts by defining a general process of interpolating noise and image:

$$\boldsymbol{x_t} = \alpha_t \boldsymbol{x_*} + \sigma_t \boldsymbol{\epsilon}. \tag{1}$$

Here, $\boldsymbol{x_*} \sim p(\boldsymbol{x})$ represents the data sampled from the data distribution, while $\boldsymbol{\epsilon} \sim \mathcal{N}(0, \boldsymbol{I})$ denotes noise drawn from a standard Gaussian distribution. $\alpha_t$ and $\sigma_t$ are two pre-defined variables used to control the interpolation trajectory and are only related to $t$. Rectified flow is a special case of the interpolation in Eq. 1 by defining $\alpha_t$ and $\sigma_t$ to change linearly with time $t$. In conditional flow matching (Lipman et al., 2022), the parameters are set as $\alpha_t = 1 - t$ and $\sigma_t = t$. In rectified flow (Liu et al., 2022), the settings are reversed, with $\alpha_t = t$ and $\sigma_t = 1 - t$.

The process of obtaining $\boldsymbol{x_t}$ can also be formulated as a probability flow ODE (PF-ODE) with a velocity field:

$$\dot{\boldsymbol{x_t}} = \boldsymbol{v}(\boldsymbol{x_t}, t), \tag{2}$$

$$\boldsymbol{v}(\boldsymbol{x_t}, t) = \dot{\alpha}_t \, \mathbb{E}[\boldsymbol{x_*} \mid \boldsymbol{x_t} = \boldsymbol{x}] + \dot{\sigma}_t \, \mathbb{E}[\boldsymbol{\epsilon} \mid \boldsymbol{x_t} = \boldsymbol{x}], \tag{3}$$

where $\dot{\alpha}_t$ and $\dot{\sigma}_t$ are the gradients of $\alpha_t$ and $\sigma_t$ with respect to $t$. We refer readers to Stochastic Interpolants (Albergo et al., 2023) and SiT (Ma et al., 2024) for detailed proofs of the above two equations.

Practically, flow-based generative models learn the velocity $\boldsymbol{v}$ by parameterizing it with neural networks using parameters $\phi$ and optimizing a flow-matching objective:

$$\min_{\phi} \int_0^1 \mathbb{E}\left[\|\boldsymbol{v_\phi}(\boldsymbol{x_t}, t) - \dot{\alpha}_t \boldsymbol{x_*} - \dot{\sigma}_t \boldsymbol{\epsilon}\|^2\right] \mathrm{d}t. \tag{4}$$

However, it is important to note that the primary objective of this work is not to sample $\boldsymbol{x_t}$ (e.g., generating images using rectified flow networks). Instead, our aim is to distill knowledge from such pretrained networks—specifically by employing the pretrained network as a loss function—to enhance the functionality of image generation networks across a wider array of applications, including image editing and text-to-3D generation.

Our proposed method is also related to widely studied diffusion-based priors. We discuss these related works in Appendix Sec E.

## 3 METHODS

In this section, we discuss the three proposed distillation methods: the RFDS (**R**ectified **F**low **D**istillation **S**ampling) baseline method, the iRFDS (inverse RFDS) method for image inversion and finally the RFDS-Rev (RFDS-Reversal) method to improve the generation quality of baseline RFDS.

**Formulation.** The problem of optimization using a pretrained rectified flow model $\boldsymbol{v_\phi}$ as loss functions can be formatted as follows: we would like to optimize $\boldsymbol{x} = g(\boldsymbol{\theta})$ by $\boldsymbol{\theta}^* = \arg\min_{\boldsymbol{\theta}} \mathcal{L}(\phi, \boldsymbol{x} = g(\boldsymbol{\theta}))$. The form of $\boldsymbol{x}$ can be Neural Radiance Field or 3D Gaussian Splatting (Kerbl et al., 2023) in the 3D generation problem, where the parameter $\boldsymbol{\theta}$ denotes the parameters of the 3D models. In the 2D case, $\boldsymbol{x}$ is directly $\boldsymbol{\theta}$. In the remainder of this section, we prove how to define the loss $\mathcal{L}$ and find its gradients with respect to $\boldsymbol{\theta}$.

### 3.1 RFDS

Consider the training loss function of flow-based generative models in Eq. 4. Using the pretrained network as a loss function can be seen as optimizing the input rather than the model parameters. Specifically, when we place the optimization variable $\boldsymbol{x}$ in the aforementioned equation, it transforms as follows:

$$\int_0^1 \mathbb{E}\left[\| \boldsymbol{v_\phi}(\boldsymbol{x_t}, t) - \dot{\alpha}_t\boldsymbol{x} - \dot{\sigma}_t\boldsymbol{\epsilon} \|^2\right] \mathrm{d}t, \text{ with } \boldsymbol{x} = g(\boldsymbol{\theta}) \text{ and } \boldsymbol{x_t} = \alpha_t\boldsymbol{x} + \sigma_t\boldsymbol{\epsilon}. \tag{5}$$

Here, the network parameter $\phi$ is fixed and $\boldsymbol{\epsilon}$ is a randomly sampled noise.

To find $\boldsymbol{\theta}$, the gradients of the above equation with respect to $\boldsymbol{\theta}$ can be written as:

$$\nabla_{\boldsymbol{\theta}}\mathcal{L}_{\text{rfds}}(\phi, \boldsymbol{x}, \boldsymbol{\epsilon}, t) = 2 \times \mathbb{E}\left[\underbrace{(\boldsymbol{v_\phi}(\boldsymbol{x_t}, t) - \dot{\alpha}_t\boldsymbol{x} - \dot{\sigma}_t\boldsymbol{\epsilon})}_{\text{Flow Residual}}(\frac{\partial(\boldsymbol{v_\phi}(\boldsymbol{x_t}, t) - \dot{\alpha}_t\boldsymbol{x} - \dot{\sigma}_t\boldsymbol{\epsilon})}{\partial\boldsymbol{x}}) \underbrace{\frac{\partial\boldsymbol{x}}{\partial\boldsymbol{\theta}}}_{\text{Generator Jacobian}}\right]. \tag{6}$$

It can be further simplified as:

$$\nabla_{\boldsymbol{\theta}}\mathcal{L}_{\text{rfds}}(\phi, \boldsymbol{x}, \boldsymbol{\epsilon}, t) = 2 \times \mathbb{E}\left[\underbrace{(\boldsymbol{v_\phi}(\boldsymbol{x_t}, t) - \dot{\alpha}_t\boldsymbol{x} - \dot{\sigma}_t\boldsymbol{\epsilon})}_{\text{Flow Residual}}(\underbrace{\frac{\partial(-\dot{\alpha}_t\boldsymbol{x} - \dot{\sigma}_t\boldsymbol{\epsilon})}{\partial\boldsymbol{x}}}_{-\dot{\alpha}_t} + \underbrace{\frac{\partial\boldsymbol{v_\phi}(\boldsymbol{x_t}, t)}{\partial\boldsymbol{x_t}}}_{\text{Network Jacobian}}\underbrace{\frac{\partial\boldsymbol{x_t}}{\partial\boldsymbol{x}}}_{\alpha_t})\underbrace{\frac{\partial\boldsymbol{x}}{\partial\boldsymbol{\theta}}}_{\text{Generator Jacobian}}\right]. \tag{7}$$

As highlighted in diffusion-based methods (Poole et al., 2022), calculating the network Jacobian is computationally expensive and reacts inadequately to low levels of noise. We observe similar behaviors in models based on flow-matching. Consequently, we also choose to disregard the rectified flow network Jacobian by setting it to the identity matrix, akin to the approach taken with diffusion priors. This modification leads to the final RFDS loss:

$$\nabla_{\boldsymbol{\theta}}\mathcal{L}_{\text{rfds}}(\phi, \boldsymbol{x}, \boldsymbol{\epsilon}, t) \simeq \mathbb{E}\left[w(t)\underbrace{(\boldsymbol{v_\phi}(\boldsymbol{x_t}, t) - \dot{\alpha}_t\boldsymbol{x} - \dot{\sigma}_t\boldsymbol{\epsilon})}_{\text{Flow Residual}}\underbrace{\frac{\partial\boldsymbol{x}}{\partial\boldsymbol{\theta}}}_{\text{Generator Jacobian}}\right]. \tag{8}$$

Here we use $w(t)$ to represent a value related to timestep $t$, and it absorbs all constant values in Eq. 7. The RFDS loss is able to use the pretrained rectified flow model as a loss function to optimize a given image $\boldsymbol{x}$. Given a rectified flow based model, we can easily obtain its corresponding RFDS loss by setting $\dot{\alpha}_t$ and $\dot{\sigma}_t$ to $-1$ and $1$ or vice versa, depending on the way it formulates Eq. 1. We include a detailed algorithm for optimization using RFDS in Appendix Sec F.

### 3.2 IRFDS

A particularly interesting characteristic of the velocity prediction objective is its time-symmetry property (Liu et al., 2022). This allows the models to flow from noise to image and also from image back to noise. Motivated by this property, we observe that our method can be easily extended to

optimize the noise. If our optimization objective changes from the input image $\boldsymbol{x} = g(\boldsymbol{\theta})$ to the noise $\boldsymbol{\epsilon}$, Eq. 7 now becomes:

$$\nabla_{\boldsymbol{\epsilon}}\mathcal{L}_{\text{irfds}}(\boldsymbol{\phi}, \boldsymbol{x}, \boldsymbol{\epsilon}, t) = 2 \times \mathbb{E}\left[\underbrace{(\boldsymbol{v}_{\boldsymbol{\phi}}(\boldsymbol{x_t}, t) - \dot{\alpha}_t\boldsymbol{x} - \dot{\sigma}_t\boldsymbol{\epsilon})}_{\text{Flow Residual}}(\underbrace{\frac{\partial(-\dot{\alpha}_t\boldsymbol{x} - \dot{\sigma}_t\boldsymbol{\epsilon})}{\partial\boldsymbol{\epsilon}}}_{-\dot{\sigma}_t} + \underbrace{\frac{\partial\boldsymbol{v}_{\boldsymbol{\phi}}(\boldsymbol{x_t}, t)}{\partial\boldsymbol{x_t}}}_{\text{Network Jacobian}}\underbrace{\frac{\partial\boldsymbol{x_t}}{\partial\boldsymbol{\epsilon}}}_{\sigma_t})\right]. \quad (9)$$

Therefore, the final form of iRFDS can be represented as:

$$\nabla_{\boldsymbol{\epsilon}}\mathcal{L}_{\text{irfds}}(\boldsymbol{\phi}, \boldsymbol{x}, \boldsymbol{\epsilon}, t) \simeq \mathbb{E}\left[w'(t)\underbrace{(\boldsymbol{v}_{\boldsymbol{\phi}}(\boldsymbol{x_t}, t) - \dot{\alpha}_t\boldsymbol{x} - \dot{\sigma}_t\boldsymbol{\epsilon})}_{\text{Flow Residual}}\right], \quad (10)$$

where the value $w'(t)$ should have opposite signs to $w(t)$ due to the relations between $\sigma_t$ and $\alpha_t$ as stated in Sec. 2.

Optimizing and recovering the original noise from an image is crucial for image inversion and editing problems. By first retrieving the original noise and subsequently modifying the caption, we demonstrate in the experiment section that our method can effectively edit real-world images. A detailed algorithm of iRFDS is included in Appendix Sec F.

### 3.3 RFDS-REV

RFDS-Reversal (RFDS-Rev) is designed to enhance the baseline RFDS in terms of generation quality. We observe that the RFDS method encounters issues similar to those found with the SDS loss, such as a lack of object details. While several approaches, like the VSD loss, have been proposed to improve the SDS loss and some are applicable to rectified flows, none shows effective results with rectified flow models based on our experiments.

---

**Algorithm 1:** The RFDS-Rev Algorithm.

1   Initialize the learnable parameter $\boldsymbol{\theta}$
2   **while** Not Converge **do**
3      Sample random timestep $t$
4      Sample random noise $\boldsymbol{\epsilon}$
5      **for** $n$ steps **do**
6         Freeze $\boldsymbol{\theta}$, optimize $\boldsymbol{\epsilon}$ with iRFDS (Eq. 10)
7      Optimize $\boldsymbol{\theta}$ with optimized $\boldsymbol{\epsilon}$ based on RFDS (Eq. 8)
8   **RETURN** $\boldsymbol{\theta}$

---

To enhance the baseline RFDS, we draw inspiration from the learning mechanisms of Reflow (Liu et al., 2022; 2024). Recall that Reflow straightens the trajectory from noise to image, leading to one-to-one matching of the image and noise. In contrast, the mode-seeking nature of the original RFDS, which optimizes the image based on random noises, leads to an averaged velocity and consequently blurred images. As shown on the left side of Fig. 2, when two noise points are sampled, although both give gradients towards the high-density areas in the image domain, the averaged gradient does not necessarily point in the correct direction. To address this issue, we first introduce a critical assumption:

**Assumption 1**: For a model finetuned with Reflow, given a sample $\boldsymbol{x}_*$ from data distribution $p(\boldsymbol{x})$, there is at most one corresponding $\boldsymbol{\epsilon}$ from noise distribution $\mathcal{N}(0, \boldsymbol{I})$ that corresponds to $\boldsymbol{x}_*$.

This assumption is straightforward, as the training process for Reflow enforces a linear, non-crossing flow (Liu et al., 2022). Based on this assumption, we propose RFDS-Reversal (RFDS-Rev), a two-stage method to improve the baseline RFDS. Specifically, as depicted on the right in Fig. 2, the first stage performs noise reversal. In this stage, we use iRFDS to perform image inversion and optimize each randomly sampled noise such that their reversal occupies a relatively static position within the noise distribution. Once the reversal is determined, in the second stage, we apply RFDS to the reversed noise. The algorithm of RFDS-Rev is summarized in Algorithm 1. In experiments, we observe that running iRFDS for just one step yields very satisfactory results. Unless specified otherwise, the experiments involving RFDS-Rev employ $n = 1$ as the default configuration. Although RFDS-Rev was initially designed with the assumption of models fine-tuned using Reflow, our experimental results reveal that it enhances the performance of the baseline RFDS on models both with and without Reflow training.

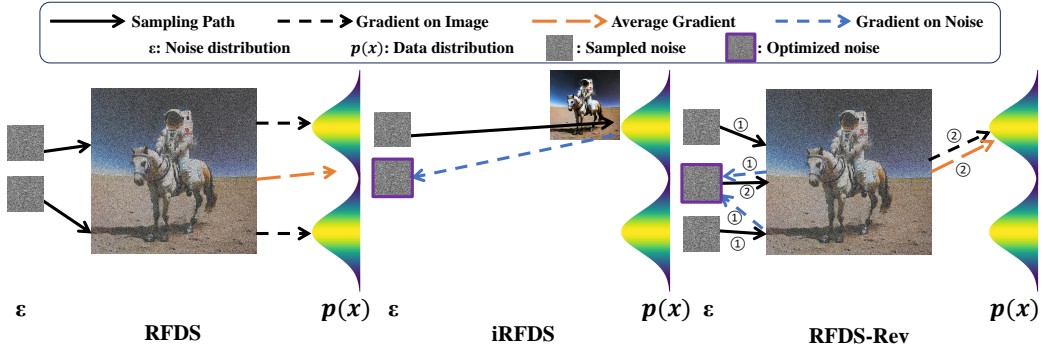

Figure 2: Illustration of our proposed methods.

**Understanding RFDS-Rev from the Angle of Euler Sampling.** In addition to the intuitive derivation above, we demonstrate that RFDS-Rev can also be mathematically derived from the difference between the RFDS baseline and the commonly used Euler sampling in rectified flow generation. Specifically, we observe that RFDS is closely related to Euler sampling, with the key difference being the noise sampling strategy. Our proposed RFDS-Rev bridges this gap by performing an inversion to recover the original noise map. The detailed proof is provided in Appendix Sec B.

## 3.4 APPLYING iRFDS AND RFDS-REV TO DIFFUSION MODELS

As discussed in previous works (Song et al., 2020; Albergo et al., 2023), diffusion models trained with score-matching objectives have a deterministic PF-ODE form and can be expressed in the form of Eq. 2. We observe that by converting the score function to a velocity field, all three of our proposed methods become applicable to diffusion models. Additionally, we prove that our proposed RFDS baseline is identical to the vanilla SDS loss when applied to diffusion models. We include a detailed proof in Appendix Sec C.

## 3.5 CLASSIFIER FREE GUIDANCE (CFG)

The above equations are derived using the rectified flow network $v_\phi$. In practice, since the base rectified flow models (Liu et al., 2024; Esser et al., 2024) are usually trained with classifier free guidance (Ho & Salimans, 2022), we observe that using a classifier free guidance version of the network $\hat{v}_\phi$ helps to improve the performance. Ablations are included in Appendix Sec G.

## 4 EXPERIMENTS

In this section, we conduct extensive experiments to compare our proposed methods with diffusion-based approaches. We perform experiments on two types of rectified flow-based methods: Stable Diffusion v3 (SD3), trained with flow-matching but without Reflow finetuning, and InstaFlow, fine-tuned with Reflow. In terms of 2D performance, Stable Diffusion v3 is one of the state-of-the-art text-to-image models, trained with more data and using larger backbones. InstaFlow achieves results similar to those of Stable Diffusion v2.1 (SD2.1) (Rombach et al., 2022). First, we focus on evaluating RFDS and RFDS-Rev in both 2D and 3D text-guided generation scenarios. Subsequently, we explore the performance of iRFDS in image inversion tasks. The diffusion-based methods are implemented using SD2.1 and the Threestudio codebase (Guo et al., 2023a). We include the implementation details in the Appendix for further reference.

## 4.1 RFDS VS. RFDS-REV VS. DIFFUSION PRIORS VS. DIFFUSION RFDS-REV

**Toy Experiments on Optimization of 2D Case.** We begin by conducting an intuitive experiment in a 2D setting, where we set $x = \theta$. This simplified setup removes the complexities of the 3D model, allowing us to demonstrate the potential upper limit of 3D generation quality. Results are shown in Fig. 3. Results reveal that our RFDS baseline, when integrated with InstaFlow, performs

| RFDS (InstaFlow) | RFDS-Rev (InstaFlow) | RFDS (SD3) | RFDS-Rev (SD3) | SDS (Diffusion) | VSD (Diffusion) | RFDS-Rev (Diffusion) |
|---|---|---|---|---|---|---|

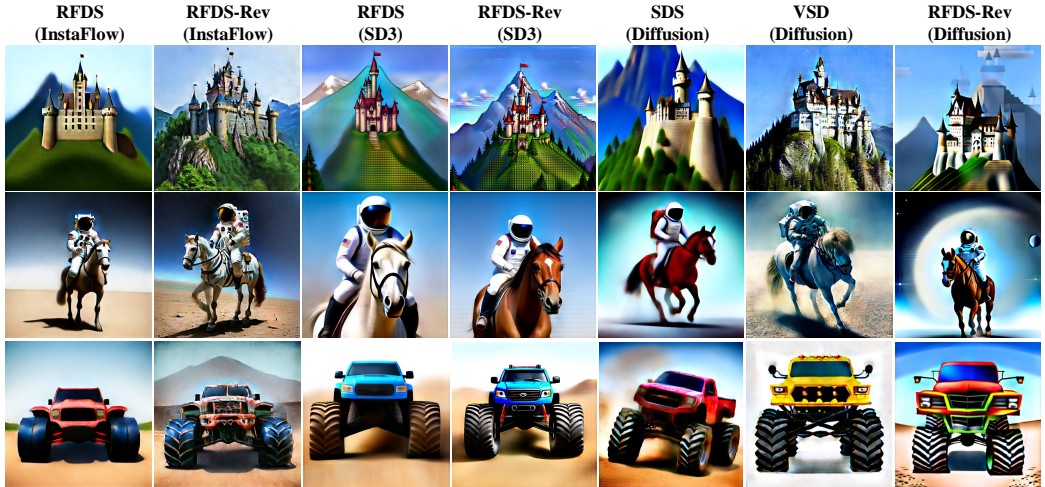

Figure 3: 2D Comparison. Text Prompts: "A castle next to a mountain", "An astronaut is riding a horse" and "A monster truck".

| RFDS (InstaFlow) | RFDS-Rev (InstaFlow) | RFDS (SD3) | RFDS-Rev (SD3) | SDS (Diffusion) | VSD (Diffusion) |
|---|---|---|---|---|---|

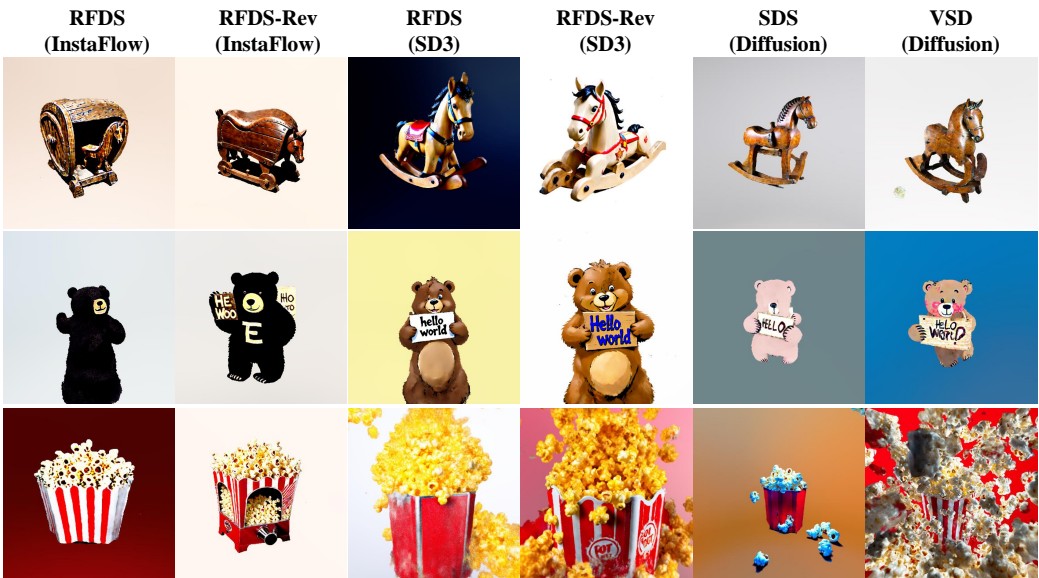

Figure 4: 3D Comparison. Text Prompts: "An antique wooden rocking horse", "A bear holding a sign that says Hello world" and "Hot popcorn jump out from the red striped popcorn maker".

comparably to the diffusion-based SDS loss. Moreover, when combined with Stable Diffusion v3, the RFDS baseline significantly enhances generation quality.

We also apply our proposed method, RFDS-Rev, to two rectified flow-based models and the diffusion model Stable Diffusion v2.1, as discussed in Sec 3.4. To better approximate the real use cases, we use only one step of iRFDS to improve efficiency for all experiments. Our results indicate that RFDS-Rev significantly enhances generation performance on InstaFlow, moderately improves performance on SD3, and slightly improves performance on the diffusion model. These findings are consistent with the assumptions made when proposing the RFDS-Rev method. The Reflow process straightens the noise-to-data trajectory, making our assumptions particularly applicable to models finetuned with Reflow. Although SD3 is not finetuned with Reflow, our method still improves generation quality. However, in score-matching models like diffusion, noise is applied stochastically to corrupt the data, making it challenging to trace the original noise with just one step of iRFDS. For

Table 1: Results of Text-to-3D on T³Bench dataset. Our proposed methods achieve state-of-the-art performance among 2D lifting methods, beating the diffusion based SDS and VSD priors.

| Method | Dreamfusion (SDS) | ProlificDreamer (VSD) | RFDS | RFDS-Rev | RFDS | RFDS-Rev |
|---|---|---|---|---|---|---|
| Base Model | Diffusion | Diffusion | InstaFlow | InstaFlow | SD3 | SD3 |
| Single Object | 24.9 | 51.1 | 46.9 | 57.6 | 49.4 | **59.8** |
| Surroundings | 19.3 | 42.5 | 34.5 | 44.6 | 39.3 | **54.5** |
| Multiple Objects | 17.3 | 45.7 | 28.3 | 44.6 | 42.9 | **55.2** |
| Average | 20.5 | 46.4 | 36.5 | 48.9 | 43.9 | **56.5** |

Figure 5: Comparison of 2D editing. Our iRFDS method is able to perform prompt-faithful editing.

this reason, in the 3D experiments, we apply our methods only to flow-matching based approaches, and compare them with SDS-like diffusion-based approaches.

**Text-to-3D Generation by Lifting 2D Models.** An important application of rectified flow or diffusion priors is lifting models from 2D to 3D. We present quantitative and qualitative results for text-to-3D generation. The qualitative results are displayed in Fig. 4. We observe that both of our proposed methods are capable of generating high-quality 3D objects, with the RFDS-Rev method enhancing the baseline RFDS in terms of object detail and color accuracy. Qualitatively, when applied to similar quality base models (InstaFlow and SD2.1), the RFDS baseline shows competitive performance compared to the SDS loss used in diffusion-based methods. Compared to VSD, RFDS-Rev achieves a comparable level of detail in generated objects. Notably, VSD experiences issues with "exploding" artifacts. In contrast, the RFDS-Rev loss does not exhibit this problem. When we apply our method to the more powerful SD3 base model, we observe that it outperforms diffusion priors. Moreover, RFDS-Rev further enhances the generation quality compared to the RFDS baseline on SD3. Additional qualitative results of RFDS-Rev are included in Appendix M.

We also conduct quantitative experiments on the text-to-3D benchmark T³Bench (He et al., 2023). The dataset contains 300 text prompts for text-to-3D generation, making it the largest text-to-3D

Table 2: Quantitative comparison of 2D editing.

| Method | DDS | Null Inversion | iRFDS (InstaFlow) | iRFDS (SD3) | iRFDS (Diffusion) |
|---|---|---|---|---|---|
| **CLIP Score** | 0.294 | 0.298 | **0.306** | 0.297 | 0.296 |
| **User Preference (Semantic)** | 7.3% | 20.1% | **43.6%** | 16.8% | 12.2% |
| **User Preference (Consistency)** | 10.7% | 32.5% | **33.4%** | 13.1% | 10.3% |

benchmark available. Results are demonstrated in Table 1. We report the quality score, derived by rendering multi-view images and calculating the CLIP (Radford et al., 2021) and ImageReward (Xu et al., 2024) scores between the rendered images and the corresponding text prompt. We observe that our proposed methods significantly outperform diffusion-based methods. Furthermore, the RFDS-Rev method sets a new state-of-the-art performance among all 2D lifting methods in this benchmark.

## 4.2 IRFDS VS. DIFFUSION METHODS

In this section, we study the performance of iRFDS in 2D image inversion and subsequent text-guided editing. Image inversion is also a widely explored topic in diffusion-based methods, known for its capability to edit real-world images. We compare iRFDS against a popular method – null inversion (Mokady et al., 2023) – on real-world images. Additionally, we evaluate our method against the diffusion prior based optimization method, the DDS loss (Hertz et al., 2023), which directly optimizes the original images.

The results of the inversion and editing experiments are summarized in Fig. 5. Among the different variants of our iRFDS methods, we find that the model finetuned with Reflow produces the best results. Further visualizations can be found in Appendix Sec. J. We also conduct quantitative experiments to calculate the CLIP score between the target caption and the image after editing. Additionally, we carry out a user study to evaluate the editing performance from the users' perspective in terms of Semantic coherence and Consistency of unrelated areas. The experiments are performed on 15 real images randomly collected from the internet, with each image being edited three times using different captions. The results are summarized in Table. 2. We observe that our proposed method iRFDS + InstaFlow achieves the best performance in terms of CLIP scores and user preference.

The above results are achieved using the simplest implementation, where the learned noise serves as the starting noise for image generation. The performance of our proposed iRFDS can be further enhanced by integrating the learned noise into an intermediate flow generation step—a simple yet effective technique commonly employed in DDIM inversion (Hertz et al., 2022) and null-inversion (Mokady et al., 2023). This method improves background consistency and ensures more accurate color reproduction. In experiments, we observed that inserting the noise at 10% of the total steps achieves the best balance between consistency and editing performance. Detailed comparisons are provided in the Appendix Sec K.

We also provide a comparison of image reconstruction ability. The experiments are conducted on the same dataset, evaluating the PSNR scores between the inversion results and the original images. Our proposed iRFDS baseline achieves a PSNR of 28.52. By applying the aforementioned insertion trick, the PSNR improves to 28.96, surpassing the null-inversion score of 28.81. A visual comparison is provided in Appendix Sec L.

## 4.3 ABLATION EXPERIMENTS

**iRFDS optimization steps in RFDS-Rev (Algorithm 1)**. Fig. 6 shows results from both the 2D and 3D cases. We observe that performing the iRFDS for 1 step achieves the best efficiency and performance trade-off.

**w/ Network Jacobian vs. w/o Network Jacobian**. As illustrated in Fig. 7, we observe that keeping the network Jacobian can hardly generate meaningful images even in the 2D case. This observation is consistent for both SD3 and InstaFlow.

**Additional Ablation Results.** We also include additional ablation results in Appendix Sec G. The experiments include: ablation on the effect of CFG, RFDS-Rev vs. directly applying the same idea of VSD to rectified flow and a comparison of computational cost between our methods and diffusion-based methods.

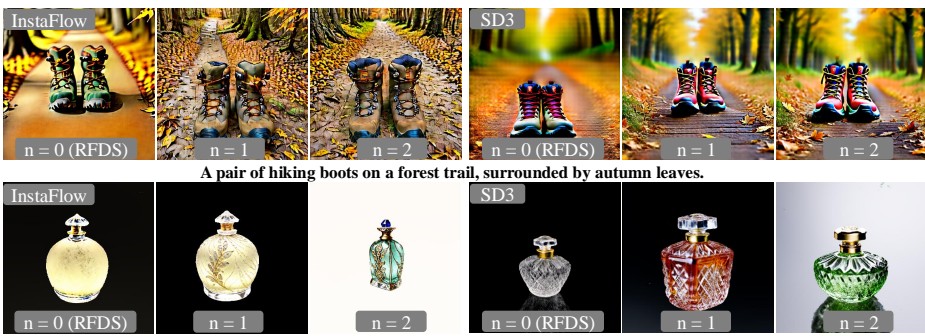

Figure 6: Experiments of the noise optimization step in RFDS-Rev. Top: 2D case. Bottom: 3D case. Optimizing the iRFDS for one step offers the optimal balance between efficiency and performance.

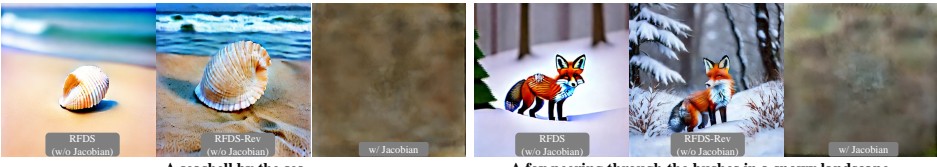

Figure 7: Ablation experiments of ignoring the rectified flow network Jacobian. We observe that it is difficult to generate images by keeping the Jacobian of the rectified flow network.

## 5  SUMMARY OF OBSERVATIONS AND DISCUSSION

**Summary of observations. 1)**, RFDS and RFDS-Rev are effective on flow-matching models, regardless of whether Reflow fine-tuning is applied (Fig. 3, 4, Table. 1). **2)**, Reflow fine-tuning enhances iRFDS performance on image inversion and editing tasks (Fig. 5, Table 2). **3)**. When applied to score-matching-based methods like diffusion models, RFDS is equivalent to the SDS loss. RFDS-Rev offers slight improvements over the SDS baseline. While iRFDS may not produce the best editing results with diffusion models, it presents a simple alternative to other inversion methods.

**Additional advantages over diffusion based priors.** Besides the performance improvements, we observe that methods based on rectified flow achieve faster convergence during the optimization of 3D models. Intuitively, if we rearrange Eq. 8, $x$ can be regarded as learning toward a direct objective $v_\phi + \epsilon$. In contrast, diffusion models rely on the difference in predicted noises to guide the optimization process. A detailed visual comparison is included in Appendix Sec H. In terms of wall-clock time, generating a single 3D scene using InstaFlow with RFDS takes less than 20 minutes on A6000 GPUs. Diffusion model-based methods, such as Stable Diffusion 2.1, utilize the same number of parameters as InstaFlow. However, the SDS loss baseline requires 40 minutes to produce a scene with reasonably good quality, while VSD takes over 1.5 hours.

## 6  CONCLUSION, LIMITATIONS

In this work, we introduce the SDS and VSD counterparts in rectified flow based methods. We propose three methods that utilize rectified flow as plug-and-play priors. Our two generative methods, RFDS and RFDS-Rev, outperform SDS and VSD in terms of text-to-3D generation. Additionally, the iRFDS method demonstrates robust capabilities in image inversion and text-guided editing.

**Limitations.** Since the 2D model lacks training with camera pose information, our proposed methods encounter similar issues to the SDS and VSD losses in 3D generation, such as the multi-face issue. These limitations can be addressed in future studies by training pose-aware models using multi-view data, similar to current diffusion based approaches (Ye et al., 2023; Shi et al., 2023a; Li et al., 2023; Shi et al., 2023b; Long et al., 2023). Moreover, currently our proposed iRFDS method does not taken the CFG mismatch issue into considerations. A promising future direction will be combining the iRFDS method with null-inversion (Mokady et al., 2023) to address the CFG issue.

ACKNOWLEDGEMENTS

This research is supported by the Agency for Science, Technology and Research (A*STAR) under its MTC Programmatic Funds (Grant No. M23L7b0021). This research is also supported by the MoE AcRF Tier 2 grant (MOE-T2EP20223-0001).

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

APPENDIX

## A  BROADER IMPACTS.

Our approach significantly expands the potential applications of pretrained rectified flow models. We demonstrate that a text-to-image model can be adapted for both image editing and text-to-3D generation tasks. However, since we utilize pretrained text-to-image rectified flow models as our foundational network, our methods might inherit the biases present in these networks.

## B  PROOF: UNDERSTANDING RFDS-REV FROM THE ANGLE OF EULER SAMPLING.

Euler sampling is one of the most fundamental and widely used sampling strategies in flow matching models (Albergo & Vanden-Eijnden, 2022; Lipman et al., 2022; Liu et al., 2024). A natural question arises: why does Euler Sampling succeed in generating high-quality 2D images, while the RFDS loss does not? Within the framework of a flow matching model, Euler sampling is defined as:

$$\boldsymbol{x_{t+\Delta t}} = \boldsymbol{x_t} + \Delta t \boldsymbol{v}(\boldsymbol{x_t}, t). \tag{11}$$

In words, the image is generated by moving a small step each time given the predicted velocity. Given the $\boldsymbol{x_t}$ predicted at each step, we can recover the original image $\boldsymbol{x_*}$ by applying the definition of $\boldsymbol{x_t}$ in Eq. 1:

$$\alpha_{t+\Delta t} \boldsymbol{x'_*} + \sigma_{t+\Delta t} \epsilon = \Delta t \boldsymbol{v}(\boldsymbol{x_t}, t) + \alpha_t \boldsymbol{x_*} + \sigma_t \epsilon. \tag{12}$$

If we re-arrange the above Equation and add $-\alpha_{t+\Delta t} \boldsymbol{x_*}$ on both side, we can have:

$$\alpha_{t+\Delta t} \boldsymbol{x'_*} - \alpha_{t+\Delta t} \boldsymbol{x_*} = \Delta t \boldsymbol{v}(\boldsymbol{x_t}, t) + \alpha_t \boldsymbol{x_*} + \sigma_t \epsilon - \sigma_{t+\Delta t} \epsilon - \alpha_{t+\Delta t} \boldsymbol{x_*}. \tag{13}$$

By dividing both side with $\Delta t$, we can have the final form of the updating rule of Euler sampler:

$$\frac{\alpha_{t+\Delta t}}{\Delta t} \Delta \boldsymbol{x_*} = \boldsymbol{v}(\boldsymbol{x_t}, t) - \dot{\alpha}_t \boldsymbol{x_*} - \dot{\sigma}_t \epsilon. \tag{14}$$

The left side of the equation indicates the direction of $\boldsymbol{x_*}$ updates. Notably, the right side of the equation is the same as the proposed RFDS loss (Eq. 8), with one key difference: in Euler Sampling, the noise $\epsilon$ is a fixed initial noise, whereas in RFDS, the noise is randomly sampled. However, in the context of 3D generation, sampling a fixed noise for the entire 3D scene is impractical because 3D optimization is inherently a stochastic process, and no fixed noise corresponds to every rendered view. Nevertheless, this "fixed noise" can be identified or learned. Our proposed RFDS-Rev addresses this by using iRFDS to perform image inversion on each rendered view, identifying the corresponding static noise $\epsilon$ and ultimately bridging the gap between RFDS and Euler sampling.

## C  PROOF: RFDS, iRFDS, RFDS-REV WITH DIFFUSION MODELS

**RFDS is Identical to SDS When Expressed in Terms of Score Function**

As proven in Stochastic Interpolants (Albergo et al., 2023), the velocity $\boldsymbol{v}(\boldsymbol{x_t})$ can be expressed in terms of a score function $\boldsymbol{s}$ learned with score-matching objective

$$\boldsymbol{v}(\boldsymbol{x_t}) = \frac{\sigma_t \boldsymbol{s}(\boldsymbol{x_t})(\dot{\alpha}_t \sigma_t - \alpha_t \dot{\sigma}_t) + \alpha_t \boldsymbol{x_t}}{\alpha_t}. \tag{15}$$

For the detailed proof of this relation, we refer the readers to Albergo et al. (2023) and Ma et al. (2024).

By substituting this relation into RFDS (Eq. 8) and considering the relation $\boldsymbol{s} = \frac{-\boldsymbol{\epsilon_\phi}}{\sigma_t}$, we directly obtain the same equation as the SDS loss

$$\nabla_{\boldsymbol{\theta}} \mathcal{L}_{\text{rfds}}(\boldsymbol{\phi}, \boldsymbol{x}, \boldsymbol{\epsilon}, t) \simeq \mathbb{E} \left[ w(t) \underbrace{(\boldsymbol{\epsilon_\phi}(\boldsymbol{x_t}) - \boldsymbol{\epsilon})}_{\text{Score Residual}} \underbrace{\frac{\partial \boldsymbol{x}}{\partial \boldsymbol{\theta}}}_{\text{Generator Jacobian}} \right]. \tag{16}$$

**iRFDS Expressed in Terms of Score Function**

Similarly, we can derive the iRFDS in terms of score function

$$\nabla_\epsilon \mathcal{L}_{\text{irfds}}(\boldsymbol{\phi}, \boldsymbol{x}, \boldsymbol{\epsilon}, t) \simeq \mathbb{E} \left[ w'(t) \underbrace{(\boldsymbol{\epsilon}_\phi(\boldsymbol{x_t}) - \boldsymbol{\epsilon})}_{\text{Score Residual}} \right]. \tag{17}$$

After the two formulas are derived, we can naturally arrive at RFDS-Rev.

## D  IMPLEMENTATION DETAILS

**RFDS and RFDS-Rev for 3D generation.**  We use the 3D model implicit model Instant-NGP (Müller et al., 2022) as the 3D backbone. Each 3D model is optimized for 15000 steps. We use a CFG of 50 for all 3D experiments and 2D toy experiments. The model is optimized with a resolution of 256 for the first 5000 steps and then 500 for the final 10000 steps. The experiments are carried out on NVIDIA A6000 GPUs. On InstaFlow, generating a single 3D scene takes approximately 30 minutes of wall-clock time with RFDS-Rev and 20 minutes with the RFDS baseline. On SD3, the same task requires around 1 hour with RFDS-Rev and 40 minutes with the RFDS baseline. We use $w(t) = 1$ on SD3 and $w(t) = -1$ on InstaFlow. Choosing iRFDS step size is important to achieve high quality generation with RFDS-Rev. For InstaFlow, we use a stepsize 1. For SD3, we observe that a stepsize of $1 - \sigma_t$ produce reasonable results.

**iRFDS for image inversion and editing.** The inversion starts from a randomly sampled Gaussian noise. We optimize the noise for 1000 steps using iRFDS and CFG 1 with a learning rate of 3*10-3. To facilitate effective noise optimization, we add one additional loss to enforce the noise follows a Gaussian distribution. Specifically, we add a loss to enforce the mean and variance of the current noise to be zero and one respectively. After the noise is optimized, we change the caption to the target caption and run the forward flow for 5 steps using CFG 1.5. We use $w'(t) = -1$ on SD3 and $w'(t) = 1$ on InstaFlow.

## E  ADDITIONAL RELATED WORKS

**Diffusion as plug-and-play priors.** Our work is greatly motivated by the development of diffusion-based priors. The earliest work of diffusion prior (Graikos et al., 2022) requires to backpropagate through the diffusion U-Net. Dreamfusion (Poole et al., 2022) recognizes that such a backpropagate process greatly hurts the performance of diffusion priors. By ignoring the U-Net Jacobian, the SDS loss proposed in Dreamfusion can be used for 3D generation and image editing. However, the initial version of the SDS loss suffers greatly from the lack of details and diversity. DDS loss (Hertz et al., 2023) proposes an improved version of the SDS to improve image editing by taking the difference between the current SDS and the source image SDS. VSD loss (Wang et al., 2023) improves the SDS loss for 3D generation problems. Specifically, it first trains a LoRA of the current 3D model and then takes the difference between the diffusion SDS and LoRA SDS. Although lots of methods have been proposed for diffusion models (Liang et al., 2024; Yang et al., 2024; Katzir et al., 2024), our work marks as the first work that studies how to effectively use rectified flow as priors.

## F  ALGORITHM FOR RFDS AND iRFDS

We list the detailed algorithm for RFDS and iRFDS in Algorithm 2 and Algorithm 3.

## G  ADDITIONAL ABLATION EXPERIMENTS

**Ablation Results on CFG.** The scale of classifier-free guidance (CFG) (Ho & Salimans, 2022) plays a crucial role in diffusion-based methods, such as SDS and VSD. We observe a very similar phenomenon with the rectified flow priors. As demonstrated in Fig. 8, both of the proposed methods require a CFG greater than 10 to learn reasonable shapes. However, when the CFG becomes excessively large, the 3D objects generated by RFDS exhibit over-saturated colors. In contrast, RFDS-Rev remains robust to large CFG values, even when the CFG exceeds 2000.

**Algorithm 2:** The RFDS Algorithm.

1 Initialize the learnable parameter $\theta$
2 **while** Not Converge **do**
3     Sample random timestep $t$
4     Sample random noise $\epsilon$
5     Optimize $\theta$ with $\epsilon$ based on RFDS (Eq. 8)
6 **RETURN** $\theta$

**Algorithm 3:** The iRFDS Algorithm.

1 Initialize the learnable parameter $\epsilon$
2 Get initial image $x$
3 **while** Not Converge **do**
4     Sample random timestep $t$
5     Optimize $\epsilon$ using fixed $x$ based on iRFDS (Eq. 10)
6 **RETURN** $\epsilon$

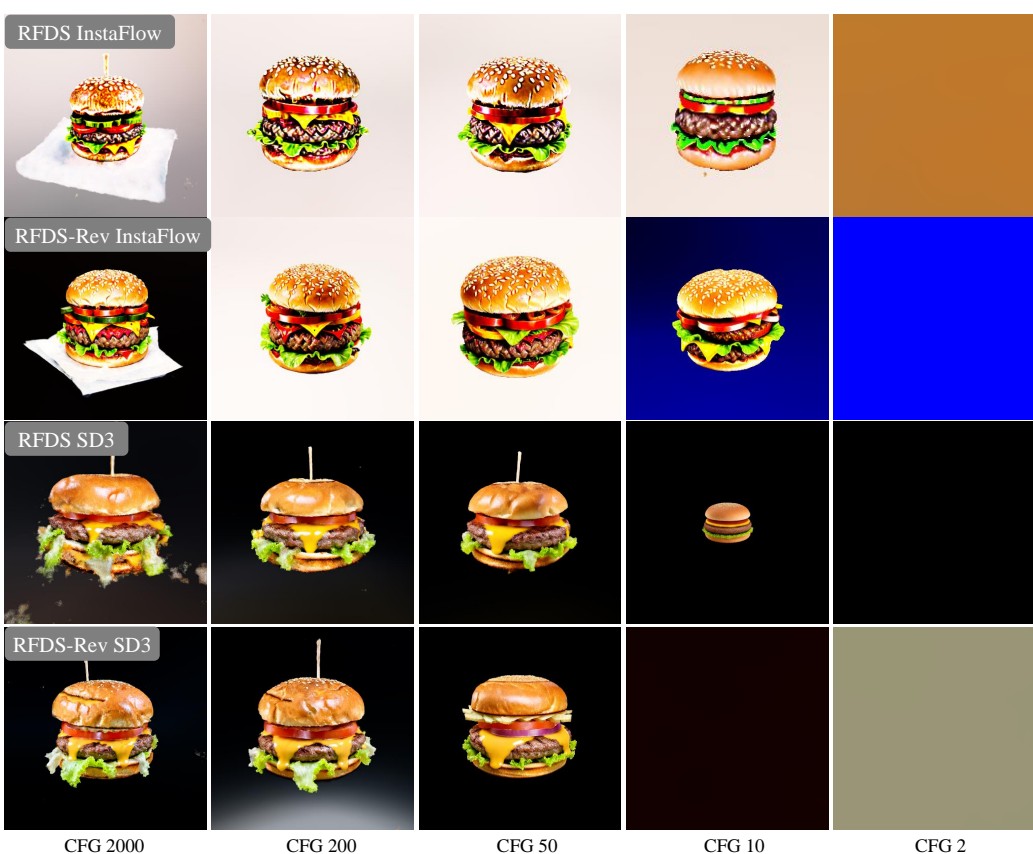

Figure 8: Ablation experiments of classifer free guidance scale on text-to-3D generation. Prompt: A DSLR image of a hamburger.

**RFDS-Rev vs. RFDS-VSD.** As mentioned in the main text, some of the existing methods aimed at improving the diffusion models can be used directly on rectified flow based methods. We explore to combine VSD with the baseline RFDS, denoted as RFDS-VSD. Specifically, we train a rectified flow LoRA model based on the current rendered images and then calculate the gradients by taking the difference between RFDS and RFDS-LoRA following the VSD setting. Results are shown in Fig. 9. Our experiments, conducted using the InstaFlow backbone, demonstrate that RFDS-Rev produces significantly better results compared to RFDS-VSD, despite RFDS-VSD requiring more computational resources. Notably, implementing VSD on the SD3 model presents significant challenges for most currently available commercial GPUs due to its requirement for fine-tuning the base model, which demands excessive GPU memory.

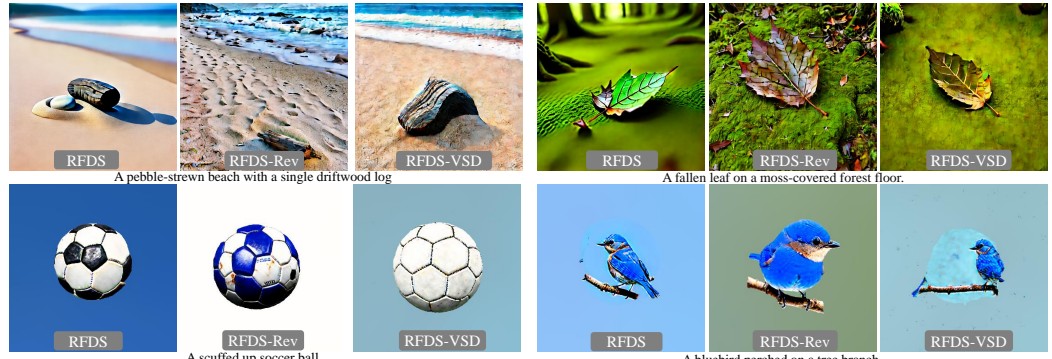

Figure 9: Ablation experiments of RFDS-Rev vs. RFDS-VSD. Top: 2D case. Bottom: 3D case.

## H    COMPARISON OF CONVERGENCE SPEED.

As listed in Fig. 10, we observe that our rectified flow based methods lead to much faster convergence speed when doing text-to-3D generation.

## I    COMPARISON OF THE COMPUTATIONAL COST.

We evaluate the computational costs by examining the number of forward and backward passes of the diffusion or rectified flow network required in 1 optimization iteration. Results are list in Table. 3. The RFDS baseline has the same computational demands as the SDS loss. Due to the calculation of CFG (Ho & Salimans, 2022), they both require two forward passes. RFDS-Rev requires only one additional forward pass, whereas VSD needs two additional forward passes and one additional costly backward pass.

Table 3: Computational cost of one iteration based on the number of forward and backward passes of the network.

| Method | | SDS Poole et al. (2022) | VSD Wang et al. (2023) | DDS Hertz et al. (2023) | **RFDS** | **iRFDS** | **RFDS-Rev** |
|---|---|---|---|---|---|---|---|
| **Category** | - | Diffusion | Diffusion | Diffusion | Rectified Flow | Rectified Flow | Rectified Flow |
| **Computation** | Forward | 2 | 4 | 2 | 2 | 1 | 3 |
| | Backward | 0 | 1 | 0 | 0 | 0 | 0 |

## J    MORE RESULTS OF IMAGE INVERSION AND EDITING USING IRFDS.

We show more results of 2D editing in Fig. 11.

## K    IMPROVE THE PERFORMANCE OF IRFDS BY INCORPORATING NOISE INTO THE INTERMEDIATE GENERATION STEP.

As discussed in the main text, the performance of our proposed iRFDS can be further enhanced by integrating the learned noise into an intermediate flow generation step. A visual comparison of this approach is presented in Fig. 12.

## L    COMPARISON OF IMAGE RECONSTRUCTION BETWEEN IRFDS AND NULL-INVERSION

A visual comparison of the image reconstruction ability between our proposed iRFDS and null-inversion is presented in Fig. 13.

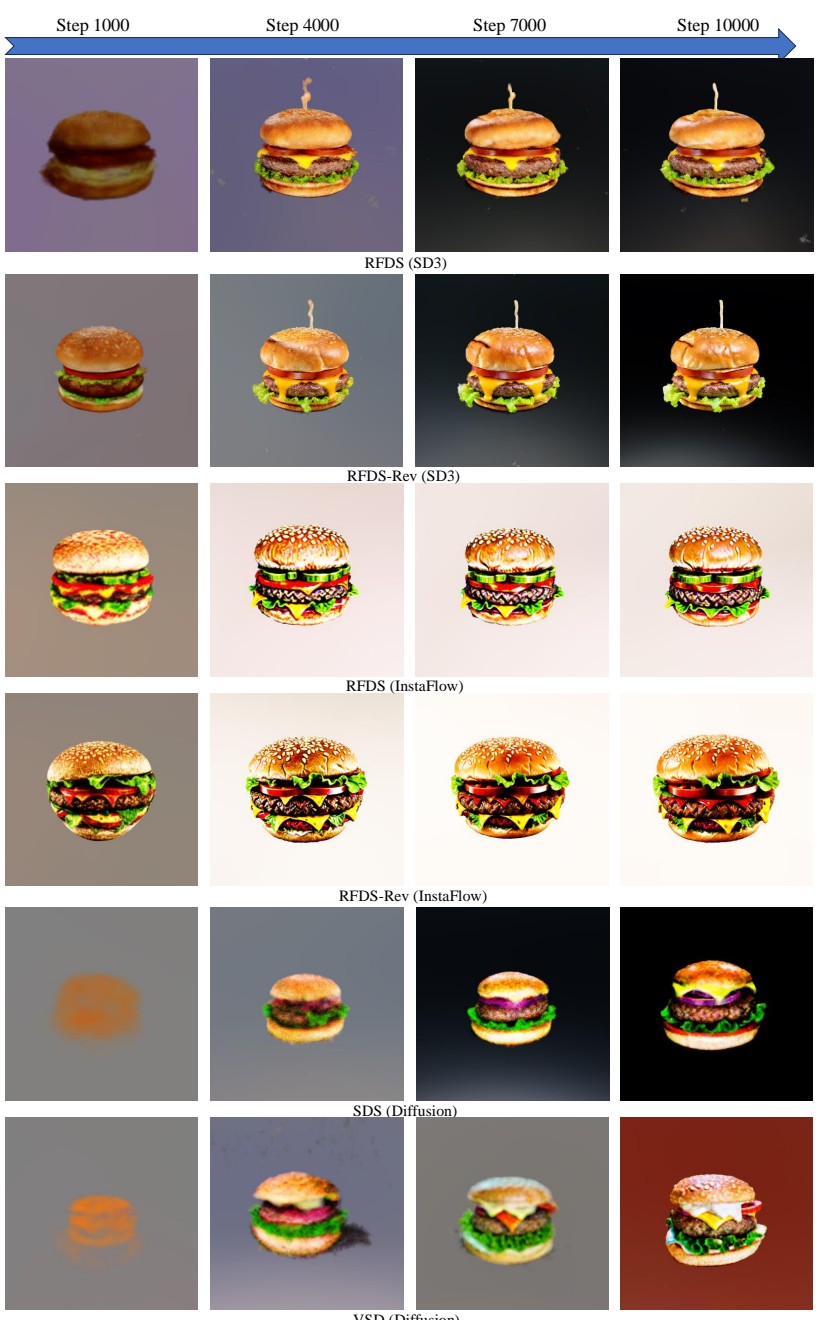

Figure 10: Comparison of convergence speed in 3D generation. Caption: A DSLR image of a hamburger. The 3D model is trained with the same learning rate. We observe that the rectified flow based methods converge much faster compared with diffusion-based methods.

## M MORE RESULTS OF TEXT-TO-3D GENERATION

We show more qualitative results of text-to-3D generation in Fig. 14,Fig. 15, Fig. 16 and Fig. 17.

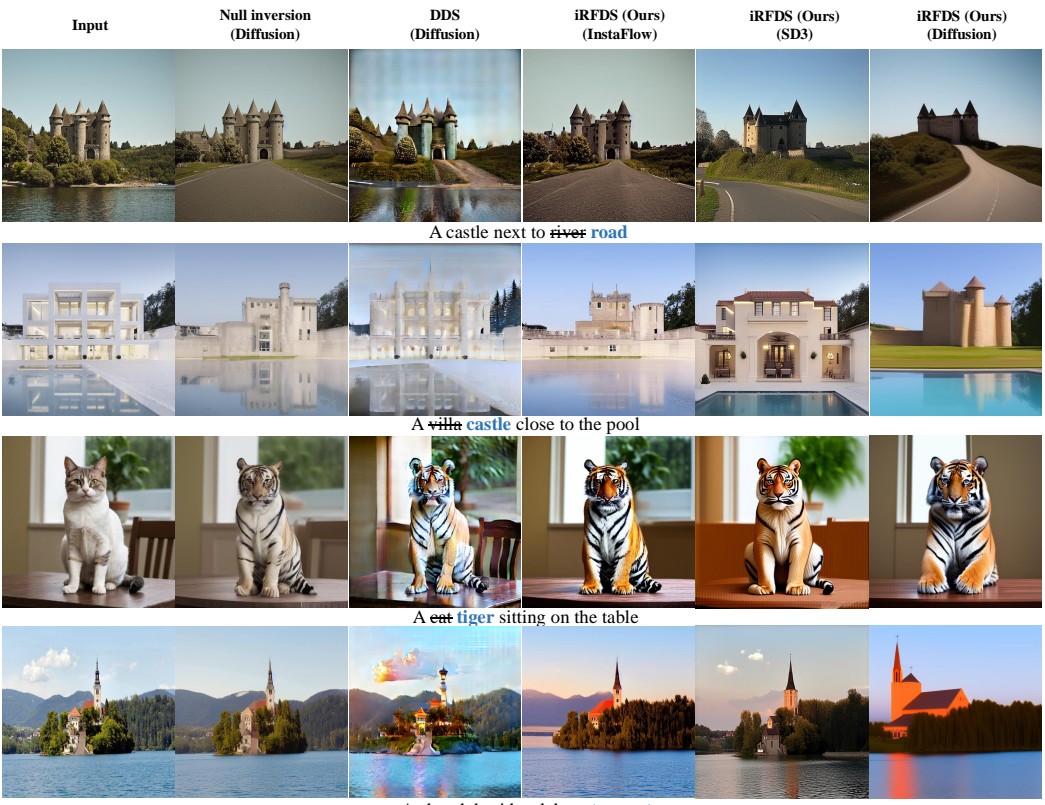

Figure 11: More results on 2D editing.

## N    iRFDS USER STUDY DETAILS

The user study is carried out with google doc. The users are ask to select the best editing results from the 4 methods. A screenshot of the user study is shown in Fig 18.

## O    MORE COMPARISON WITH OTHER STATE-OF-THE-ART 3D GENERATION METHODS

We further compare our proposed method (RFDS-Rev + SD3) with other state-of-the-art 3D generation methods, including LucidDreamer (Liang et al., 2024), DreamCraft3D (Sun et al., 2023), and GaussianDreamer (Yi et al., 2023). The results are presented in Fig 19. Our findings demonstrate that our proposed method is versatile, as it can be applied to both NeRF and 3DGS backbones. Our method achieves highly competitive performance across different settings.

| Original Image | iRFDS (Step 0) | iRFDS (Step 0.1) | Null-inversion |
|---|---|---|---|

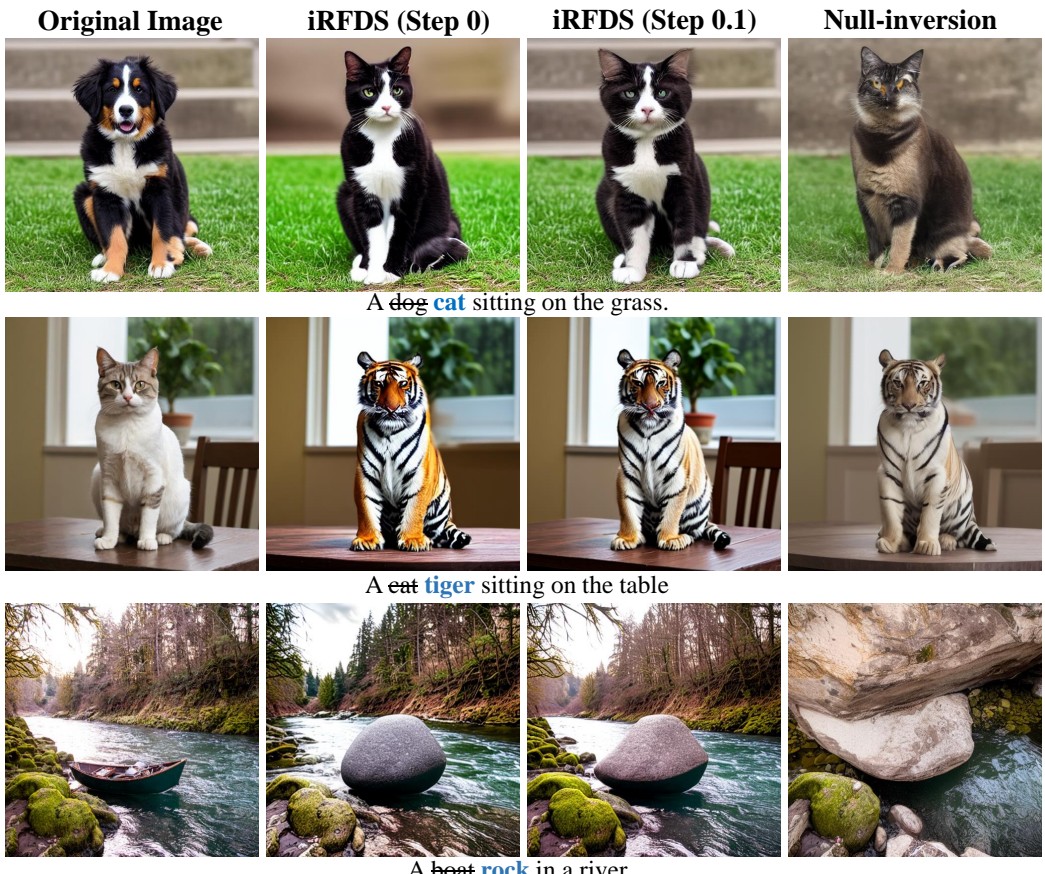

A ~~dog~~ **cat** sitting on the grass.

A ~~cat~~ **tiger** sitting on the table

A ~~boat~~ **rock** in a river

Figure 12: Inserting the iRFDS learned noise into the intermediate generation step improves background and color consistency.

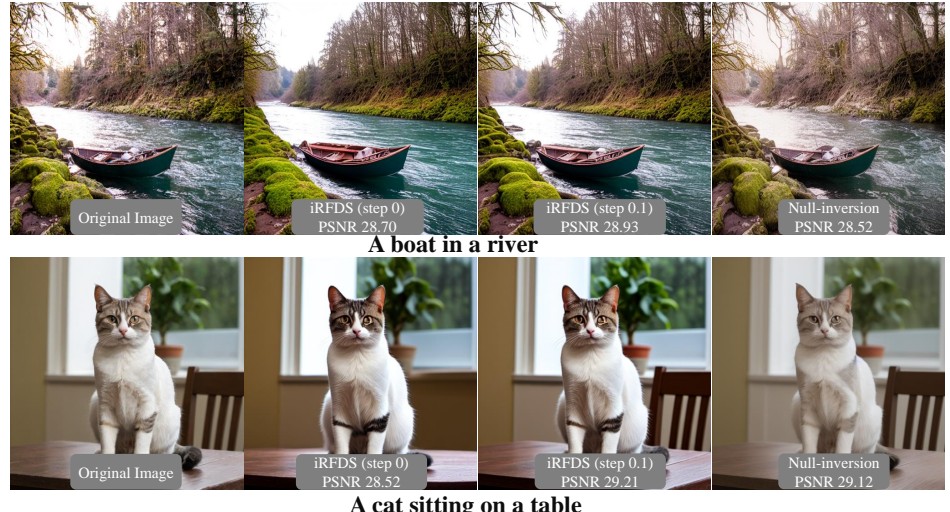

A boat in a river

A cat sitting on a table

Figure 13: Comparison of image reconstruction.

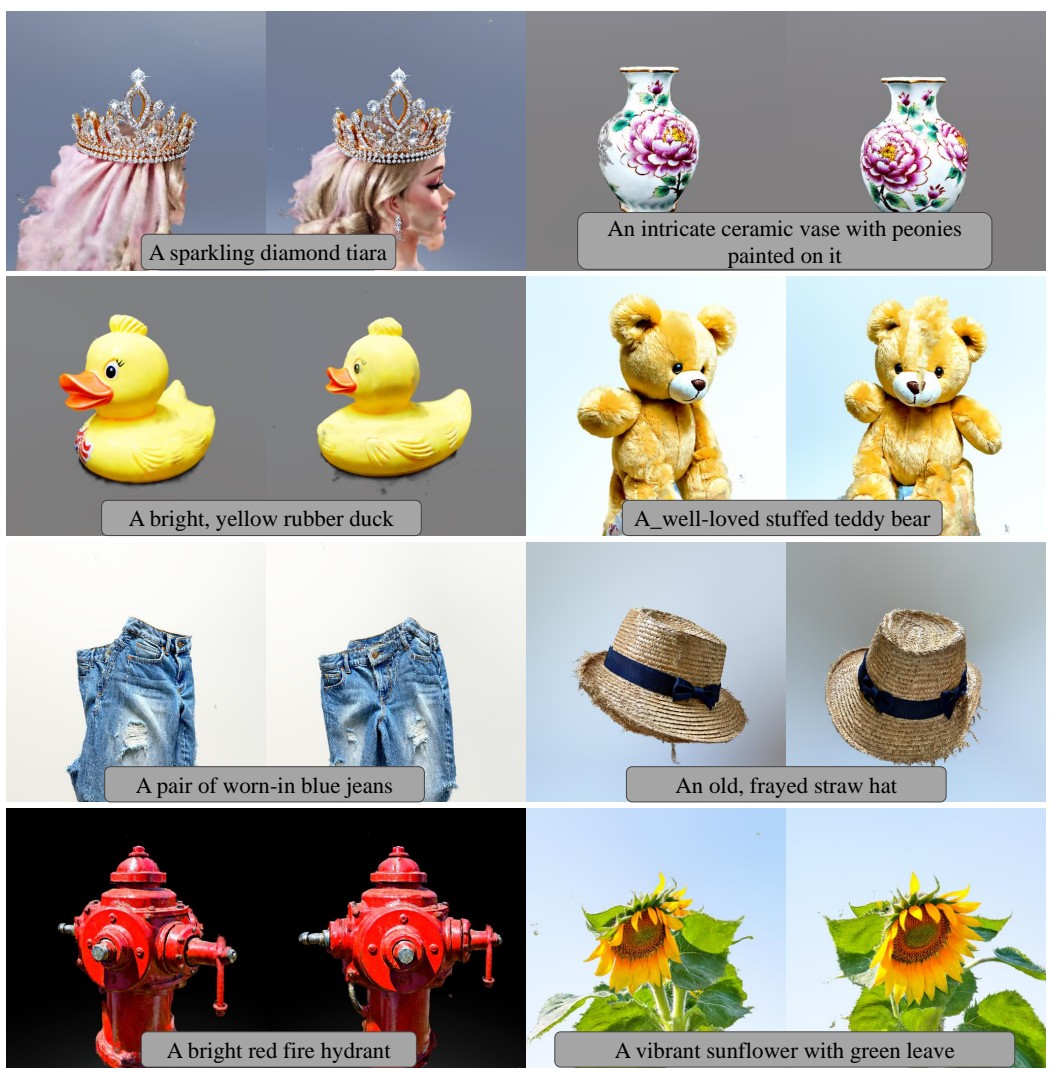

Text-to-3D Generation with RFDS-Rev

Figure 14: More results on text-to-3D generation. Model:SD3

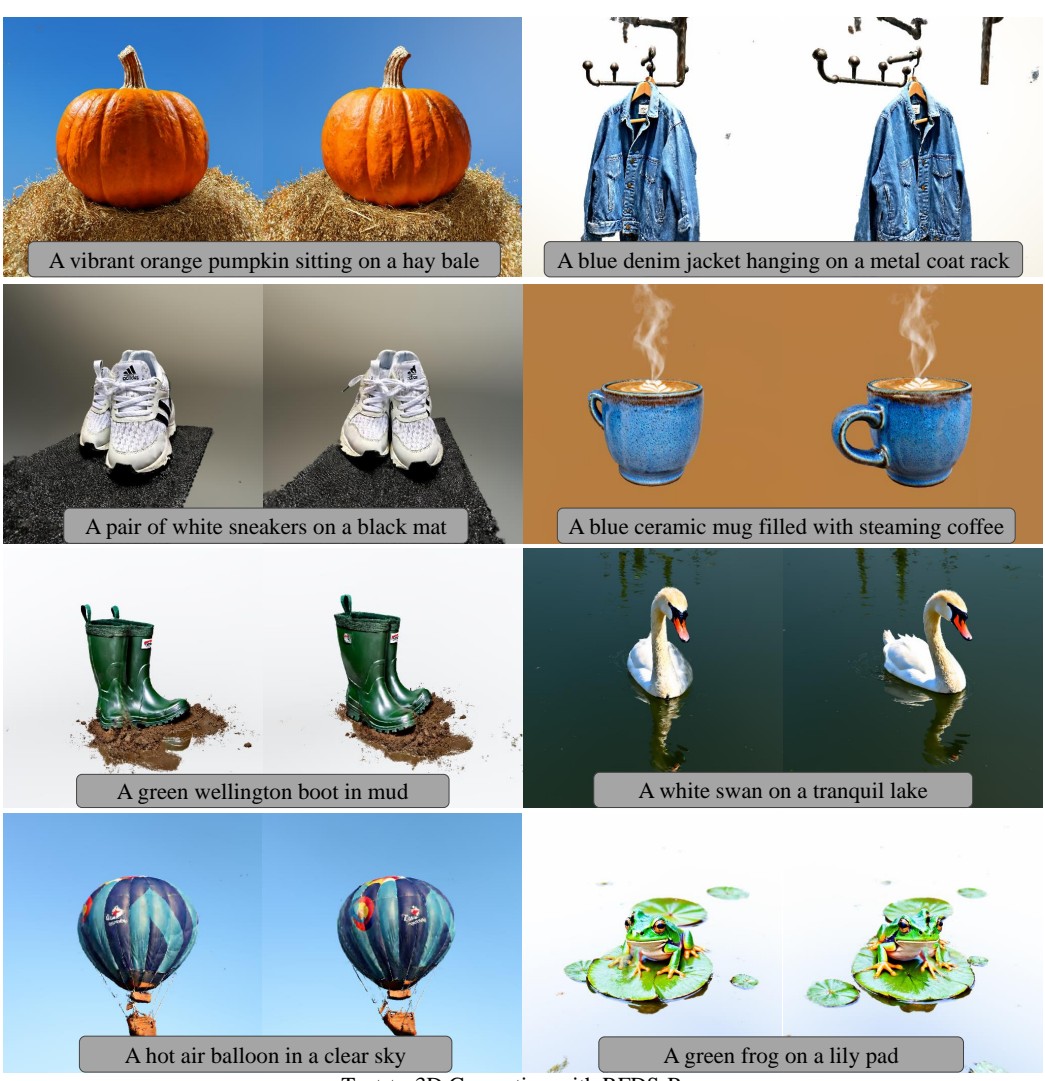

A vibrant orange pumpkin sitting on a hay bale

A blue denim jacket hanging on a metal coat rack

A pair of white sneakers on a black mat

A blue ceramic mug filled with steaming coffee

A green wellington boot in mud

A white swan on a tranquil lake

A hot air balloon in a clear sky

A green frog on a lily pad

Text-to-3D Generation with RFDS-Rev

Figure 15: More results on text-to-3D generation. Model:SD3

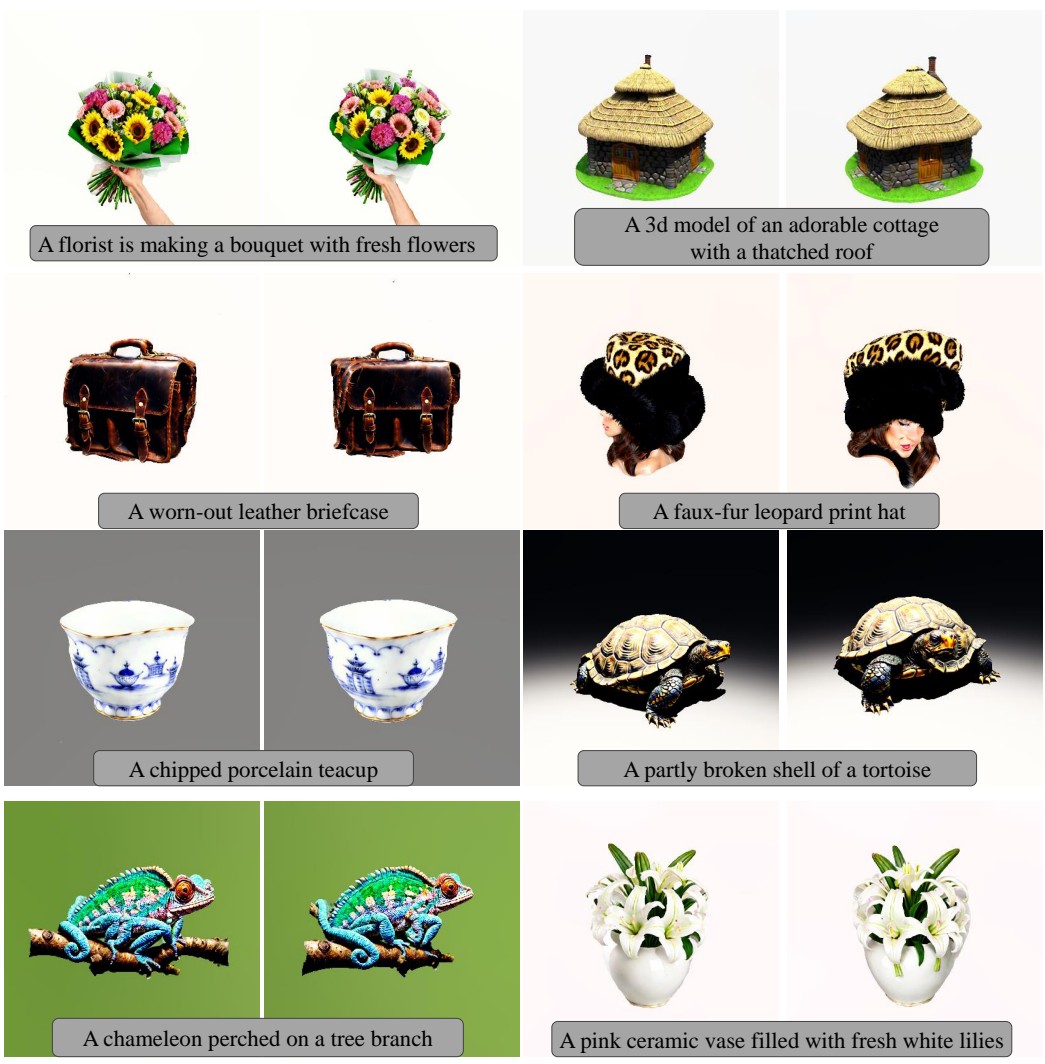

Text-to-3D Generation with RFDS-Rev

Figure 16: More results on text-to-3D generation. Model:InstaFlow

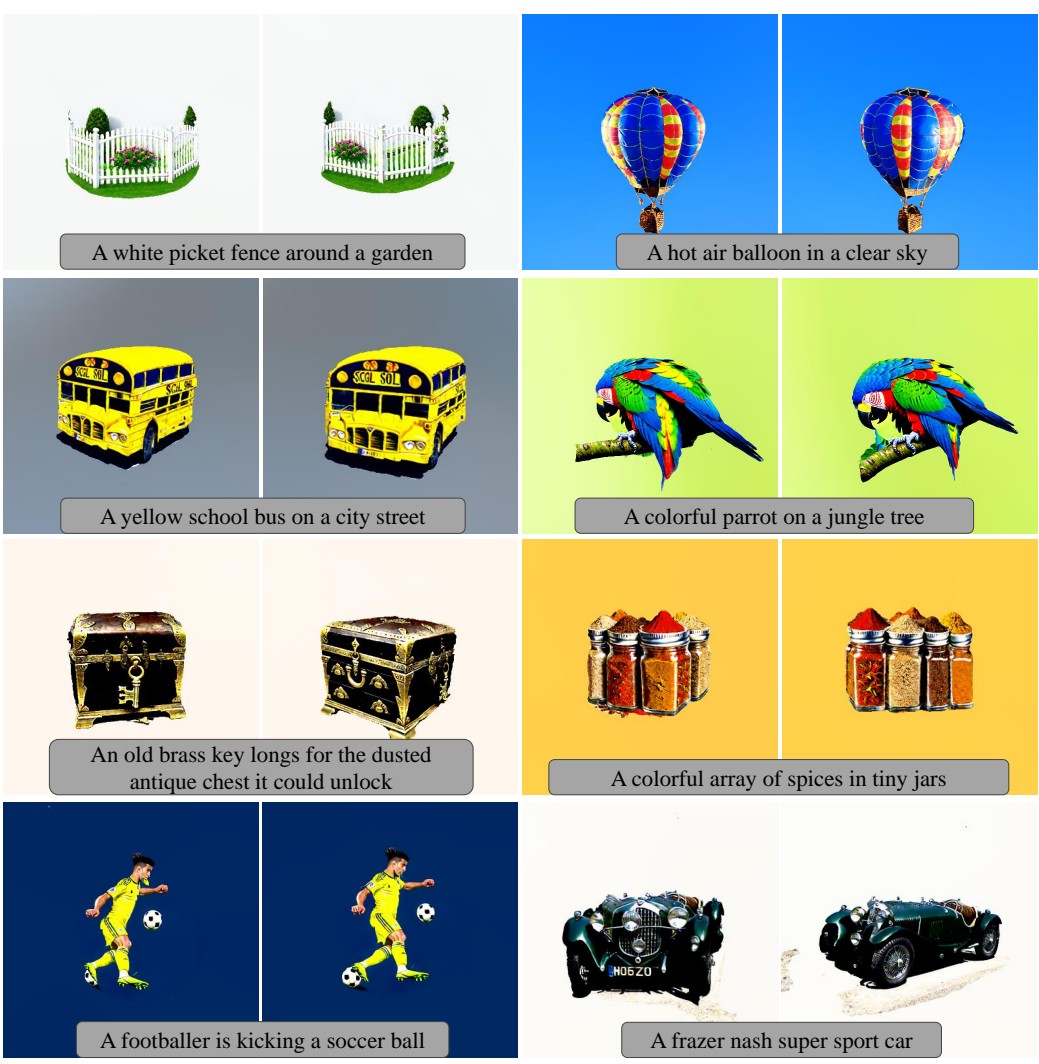

Text-to-3D Generation with RFDS-Rev

Figure 17: More results on text-to-3D generation. Model:InstaFlow

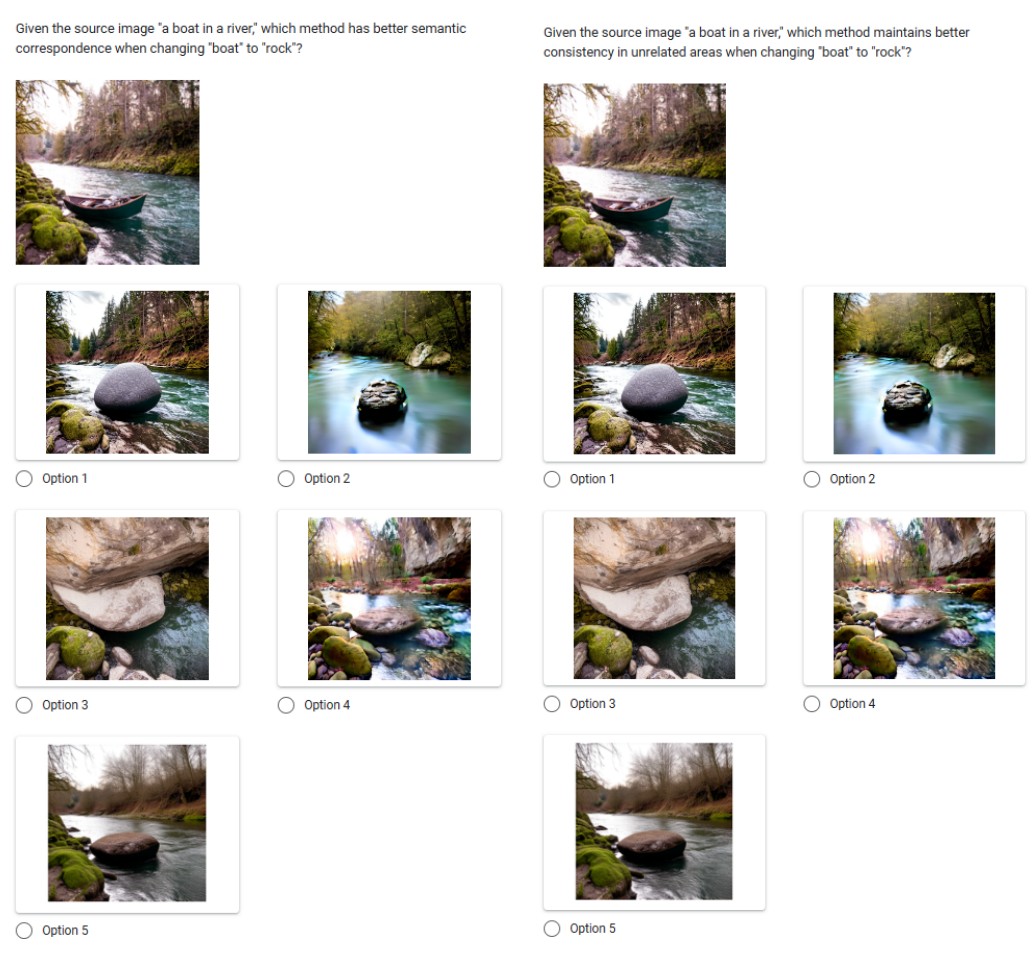

Figure 18: User study page

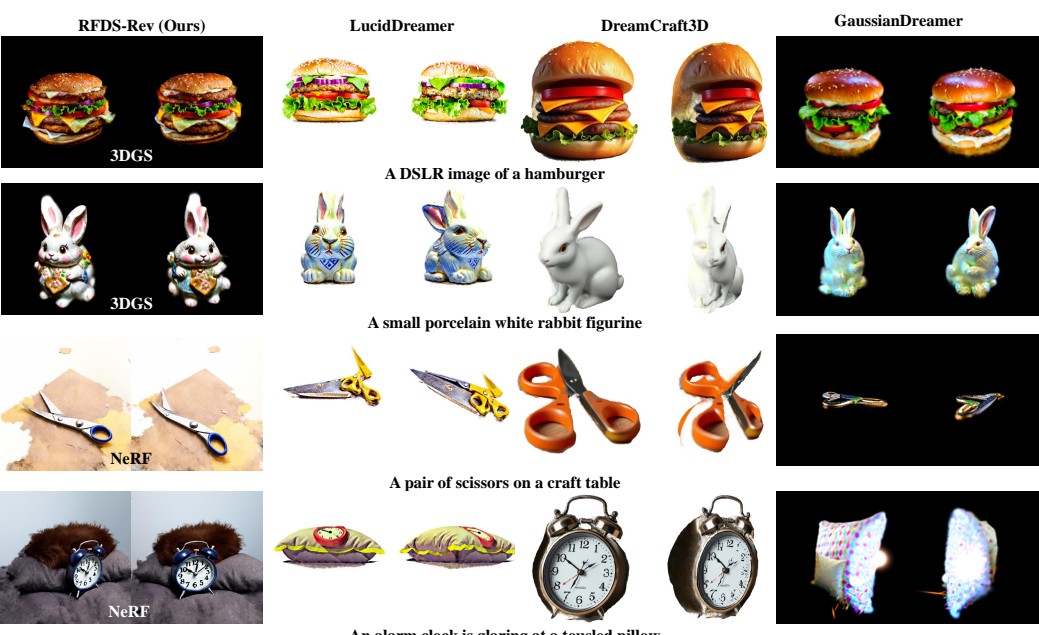

Figure 19: Comparison with other state-of-the-art 3D generation methods

