# OpenReview forum: "Text-to-Image Rectified Flow as Plug-and-Play Priors"
_ICLR.cc/2025/Conference — ICLR 2025 Poster_

### Official Review · Reviewer_6eY8 · 2024-10-29

**Soundness:** 2
**Presentation:** 3
**Contribution:** 3
**Rating:** 6
**Confidence:** 3

**Summary:**

This paper presents a framework that distills knowledge from the pre-trained text-to-image rectified flow models, which serve as a strong prior and apply to various downstream tasks such as image editing and text-to-3D. Similar to ‘Score Distillation Sampling’ (SDS) loss, authors propose a novel loss termed ‘Rectified Flow Distillation Sampling’ (RFDS) that is fully compatible with Rectified Flow models and partially with Diffusion models. Furthermore, they suggest iRFDS and RFDS-Reversal, the variant of their proposed loss, that can be applied to the image inversion and RFDS-based enhancement mechanism, respectively. The authors conducted extensive experiments to validate the efficiency and efficacy of their method.

**Strengths:**

1. The proposed RFDS, which poses similarity to SDS, presents an effective way to adapt pre-trained Rectified Flow models to tasks such as text-to-3D. This contribution is fundamental since there is a growing trend toward Rectified Flow models in the community.

2. The authors conducted extensive experiments with various Rectified Flow/Diffusion baselines, which proves the versatility and efficacy of the proposed RFDS framework.

3. The proposed method shows solid performance and faster convergence speed compared to baselines, which is noticeable.

4. This paper is easy to follow and comprehend.

**Weaknesses:**

1. As far as my understanding is correct, RFDS-Rev redirects the gradients (velocities) of the rendered image to points close to the same data distribution mode given different noisy samples to handle common issues such as blurriness that could arise from averaging different-mode directing gradients. For this reviewer, it is unclear how iRFDS can enhance the performance of models without Reflow since these models would not have perfect straight-line paths. Approximated starting points (initial noise) would not guarantee a ‘static position’ within the noise distribution, and I think the more detailed explanation or empirical evidence would further make this work concrete.

2. The results in image editing (Fig.5) do not seem convincing since semantics that are supposed to remain still are also being changed, such as the global contrast of the image, textures, and object shapes. This makes me doubtful if the proposed iRFDS can indeed optimize the valid initial noise given data. Maybe a simple reconstruction test (from optimized noise to the original data) would further provide a concrete evaluation.

3. I think the explanation of faster convergence speed compared to SDS/VSD is weak since it is an important property in the field of 3D generation methods using diffusion priors. Further elaboration or evidence of this feature would help readers fully comprehend their work (e.g., comparing convergence curves with respect to the number of iterations or wall-clock time).

4. Weak baseline methods. While I appreciate the paper's contribution (the first attempt to use Rectified Flow as priors) and the results are quite promising, baseline methods are rather weak (DreamFusion and VSD are quite outdated at the moment). It would be great if the authors compared against state-of-the-art text-to-3D methods, e.g., LucidDreamer [1], DreamCraft3D [2] or even GaussianDreamer [3] using different 3D representations.

[1] LucidDreamer: Towards High-Fidelity Text-to-3D Generation via Interval Score Matching, Liang et al., CVPR 2024

[2] DreamCraft3D: Hierarchical 3D Generation with Bootstrapped Diffusion Prior, Sun et al., ICLR 2024

[3] Fast Generation from Text to 3D Gaussians by Bridging 2D and 3D Diffusion Models, Yi et al., CVPR 2024

**Questions:**

Please see the weakness section.

---

> ### Author Response · Authors · 2024-11-21
> **Authors Response**
>
> Thank you for the insightful feedback and recommendations!
>
> **Q1 it is unclear how iRFDS can enhance the performance of models without Reflow since these models would not have perfect straight-line paths.  more detailed explanation or empirical evidence would further make this work concrete.**
>
> **A1:** Thank you for your question. In the revision, we provide a more detailed mathematical justification of RFDS-Rev from the perspective of Euler sampling. The details are given in Lines 265–269, with the full proof in Appendix Sec. B (Page 14). Specifically, we show that the RFDS baseline is closely related to Euler sampling in image generation. The key difference is that RFDS uses random noise, while Euler sampling uses a fixed initial noise. Our proposed RFDS-Rev bridges this gap by learning to map the image back to the initial noise.
>
> For models without reflow, it is indeed challenging to fully recover the noise. However, we hypothesize that, while the original noise cannot be fully and accurately recovered, the iRFDS operation helps reduce the noise variance compared to previous random sampling. This is supported by the results shown in Table 1, Fig. 3, and Fig. 4.
>
> **Q2 The results in image editing (Fig.5) do not seem convincing since semantics that are supposed to remain still are also being changed**
>
> **A2:** Thank you for your suggestion.
>
> **Reasons for background changing compared with null-inversion**: In our original submission, we used the simplest implementation of the iRFDS loss without incorporating any specific tricks. This issue can be largely resolved by adding a small, commonly used trick from DDIM inversion—injecting the learned noise into an intermediate step.
>
> In the revision, we address this problem in Lines 455–461 and provide a comparison in Appendix Sec. K and Fig. 12 (Page 20). Our results show that the saturation and background-changing issues are significantly mitigated by using this simple trick.
>
> **Reconstruction test** We conducted a reconstruction test as requested in Lines 463–467. Our observations indicate that by applying the same small trick used in null-inversion, our proposed iRFDS achieves a higher PSNR (28.96) compared to null-inversion (28.81).
>
> We also provide a visual comparison in Fig. 13 (Page 20) to further illustrate the improvements.
>
> **Q3 I think the explanation of faster convergence speed compared to SDS/VSD is weak**
>
> **A3:** Thank you for your suggestion. We have included the visual results in Fig. 10 (Page 18). In the revision, we also added a comparison of wall-clock time in Line 521:
>
> "In terms of wall-clock time, generating a single 3D scene using InstaFlow with RFDS takes less than 20 minutes on A6000 GPUs. Diffusion model-based methods, such as Stable Diffusion 2.1, utilize the same number of parameters as InstaFlow. However, the SDS loss baseline requires 40 minutes to produce a scene with reasonably good quality, while VSD takes over 1.5 hours."
>
>
> **Q4 Weak baseline methods. Comparison with LucidDreamer, DreamCraft3D, GaussianDreamer**
>
> **A4:**
> Thank you for your suggestion. The main contribution of this paper is the proposal of a flow prior, which is why, in our initial experiments, we focused on comparing our method against other diffusion priors.
>
> On the other hand, LucidDreamer and GaussianDreamer focus on backbone side improvement and Shape E for shape initialization. While LucidDreamer also introduces an improved diffusion prior, it requires a multi-step forward process, making it too time-consuming to integrate into large models like SD3. DreamCraft3D, on the other hand, utilizes a pose-finetuned diffusion model and external normal estimation to improve performance.
>
> However, we agree with your point that a comparison with these methods would be valuable. Therefore, we have included a side-by-side comparison in the Appendix Sec O (page 19) to compare our proposed methods with those mentioned.

---

> ### Author Response · Authors · 2024-11-26
> **Comment by Authors**
>
> Dear Reviewer,
>
> We hope our reply could address your questions.
>
> As the discussion phase is nearing its end, we would be grateful to hear your feedback and wondered if you might still have any other concerns we could address.
>
> We thank you again for your effort in reviewing our paper!
>
> Authors

---

> > ### Comment · Reviewer_6eY8 · 2024-11-27
> >
> > Thank you for addressing the majority of the concerns I raised. I am still unclear about how the proposed approach achieves faster convergence. Additionally, I believe the paper would benefit from quantitative comparisons with the state-of-the-art baselines. However, I understand the constraints of the revision timeline. Overall, while the work is competent, the technical contribution does not strike me as particularly significant. Therefore, I maintain my score as a borderline accept.

---

> > > ### Author Response · Authors · 2024-12-03
> > > **Comment by Authors**
> > >
> > > Dear Reviewer 6eY8,
> > >
> > > Thank you very much for your positive feedback and encouraging rating!
> > >
> > > **Regarding how the proposed approach achieves faster convergence:**
> > >
> > > As noted in Lines 515–523 of the manuscript, the observed faster convergence is currently an empirical finding based on our experimental results.
> > > To provide further clarity, we have included a visual comparison in Appendix Section H and Fig. 10 (page 18).
> > >
> > > Since this paper represents the first exploration of rectified flow as priors, we believe that an in-depth technical explanation of this phenomenon would be a valuable direction for future research.
> > >
> > > Best regards,
> > >
> > > Authors

---

### Official Review · Reviewer_6Xia · 2024-11-03

**Soundness:** 3
**Presentation:** 2
**Contribution:** 2
**Rating:** 5
**Confidence:** 3

**Summary:**

This paper uses the SOTA rectified-flow-based model as plug-and-play priors to lift the text-to-2D model into a text-to-3D model, with RFDS and RFDS-Rev. By removing the Jacobian of the rectified flow network, RFDS can generate meaningful images or 3D objects given text conditions. RFDS-Rev iteratively applies iRFDS for flow reversal to determine the original noise, and RFDS for knowledge distillation to refine the input.

**Strengths:**

1. The paper analyzes the refined process of the rectified flow.

2. Using the rectified flow as the priors is interesting.

3. The experiments are sufficient.

**Weaknesses:**

* Writing needed to be improved, especially, from Lines 100-107, which is important but the logic is somewhat unclear.

* The focus of the paper is a little confusing, including Image inversion, editing, and text-to-3D generation. In my view, text-to-3d must be the key contribution as it use the 2D model as the priors.

* What is the difference between RFDS Loss and SDS loss. It seems that RFDS is the version of the flow-based model.

* What is the speed impact of the iterative application of iRED in REDS-rev?

**Questions:**

See weaknesses

---

> ### Author Response · Authors · 2024-11-21
> **Authors Response**
>
> Thank you for the critical comments and insightful feedback.
>
> **Q1 Writing needed to be improved, especially, from Lines 100-107**
>
> **A1:**
> Thank you for your suggestion. As also recommended by reviewer cYSo in Q6, we have rewritten the introduction section in the revision.
> For the part you mentioned, instead of providing all the details in the introduction, we removed the verbose descriptions and instead provided an outline for RFDS-Rev.
> The revised content can be found in Lines 95-101.
>
> **Q2 The focus of the paper is a little confusing, including Image inversion, editing, and text-to-3D generation. In my view, text-to-3d must be the key contribution as it use the 2D model as the priors.**
>
> **A2:**
> Thank you for your question. We would like to clarify that both 3D generation and image inversion/editing are common tasks when using diffusion and flow as priors. DDS (Hertz et al., 2023) is one example of using a SDS-like method for image editing.
> Our method is versatile and can handle both tasks effectively.
>
> **Q3 What is the difference between RFDS Loss and SDS loss. It seems that RFDS is the version of the flow-based model.**
>
> **A3:** Thank you for your question. RFDS is the baseline loss we derived from flow-matching based models and the SDS loss is for diffusion models. Moreover, the relationship between the RFDS and SDS is addressed in Sec. 3.4 and formally proven in Appendix Sec. C.
> We demonstrate that by considering the ODE form of the diffusion SDE and converting the velocity field to score functions, our proposed RFDS baseline is mathematically identical to the vanilla SDS loss when applied to diffusion models.
>
> In terms of our contributions, RFDS serves as a straightforward baseline derived from flow-matching methods. Our contributions also include the proposal of iRFDS for image inversion and editing, as well as RFDS-Rev, which enhances the RFDS baseline by addressing its limitations and improving overall performance.
>
> **Q4 What is the speed impact of the iterative application of iRED in REDS-rev?**
>
> **A4:**
>
> In our original submission, we discussed this issue in Appendix Sec. D and Sec. I. In our experiments, we only run iRFDS one time each step. Therefore, RFDS-Rev requires only one additional forward pass compared to the RFDS baseline. In terms of wall-clock time, generating a single scene with RFDS-Rev takes 30 minutes, while RFDS takes 20 minutes.

---

> > ### Comment · Reviewer_6Xia · 2024-11-21
> >
> > Thank the authors. All my issues are addressed. I increase the score to 5.

---

### Official Review · Reviewer_cYSo · 2024-11-04

**Soundness:** 3
**Presentation:** 3
**Contribution:** 3
**Rating:** 6
**Confidence:** 5

**Summary:**

This paper proposes to use rectified flow for SDS instead of diffusion models.

The propose method has three components:
* Rectified flow distillation sampling (RFDS) = typical SDS but with rectified flow
    * Equations are the same with SDS except having *flow residual* instead of denoising loss
    * It uses random noises to compute the *flow residual*.
* Optimizing the *noise* (iRFDS) with the same flow residual
* Full algorithm (RFDS-Rev) with RFDS and iRFDS
    * iRFDS starting from a random noise
    * RFDS with the optimized noise

iRFDS and RFDS-Rev are applicable to diffusion models, especially with classifier-free guidance.

**Strengths:**

1. This paper tackles a long-standing problem: SDS. SDS with rectified flow is not good enough and the proposed method generates sharp results.
2. The proposed method is easy to understand and mostly sound.
3. The proposed method is generalizable to a wide range of flow-based methods.
4. Preliminary is thorough enough to provide the knowledge base.
5. Figure 2 greatly helps understanding the intuition of RFDS-Rev.
6. Experiments are well-organized from 2D to 3D.

**Weaknesses:**

(Ordered by importance. Resolving them will raise my rating.)

1. Subsection 3.2 should provide the theoretical justification for the reason why optimizing the noise helps RFDS.
2. Choice of the competitors for text-based image editing is not sound because it covers only inversion variants. Answering following questions may improve soundness: Why should we compare only with inversion variants? Why prompt-to-prompt variants (e.g., DDPM inversion + P2P) should be ignored?
3. The text-to-3D results still suffer from Janus problem. Discussion in this direction would enrich the paper.
4. Question of the user study is ambiguous: Given the source image, "a boat in a river", which method is better by changing boat -> rock? There are different aspects of being "better", e.g., consistency except the boat, and they should be separately evaluated.
5. Results of generated 3D assets (Figure 4) should be rendered in multiple views to be evaluated. Supp. materials have viewpoint-varying videos. Mentioning a Supp. would suffice.
6. Introduction is verbose. It would be clearer if non-essential sentences are removed.

Misc.

L169 using pretrained rectified flow models v_phi -> using a pre-trained rectified flow model v_phi

**Questions:**

1. Why do the bottles change across n=1 and n‎ = 2 while boots stay the same in Figure 6? The only difference I catch between boots and bottles is 2D and 3D.
2. Why does the network Jacobian harm the results?

---

> ### Author Response · Authors · 2024-11-21
> **Authors Response**
>
> Thank you for the great paper summary, insightful comments and constructive feedback!
>
> **Q1. Subsection 3.2 should provide the theoretical justification for the reason why optimizing the noise helps RFDS.**
>
> **A1:** Thank you for the suggestion—it has helped make our paper more comprehensive. In the original Section 3.3, we derived the RFDS-Rev algorithm from an intuitive perspective.
>
> In the revision, we complement this with a mathematical justification based on Euler sampling, detailed in Lines 265–269, with the proof provided in Appendix Sec. B (Page 14). Specifically, we demonstrate that the RFDS baseline is closely related to Euler sampling in image generation, with the key difference being that RFDS utilizes random noise, while Euler sampling employs a fixed initial noise.
>
> Our proposed RFDS-Rev bridges this gap by learning to map the image back to the initial noise, thereby unifying these approaches.
>
> **Q2. Choice of the competitors for text-based image editing. Why prompt-to-prompt variants (e.g., DDPM inversion + P2P) should be ignored?**
>
> **A2:**
> Thank you for your question. In our experiments, we utilized null-inversion, which is an enhanced version of DDIM inversion with p2p support.
>
> Compared to DDIM inversion, null-inversion specifically aims to address the CFG mismatch issue of DDIM inversion and p2p, offering improved performance and consistency compared with the DDIM inversion baseline.
>
> **Q3.The text-to-3D results still suffer from Janus problem. Discussion in this direction would enrich the paper.**
>
> **A3:** Thank you for your question. In our original submission, we addressed this issue in the limitations section. To quote:
>
> "Since the 2D model lacks training with camera pose information, our proposed meth-
> ods encounter similar issues to the SDS and VSD losses in 3D generation, such as the multi-face
> issue. These limitations can be addressed in future studies by training pose-aware models using
> multi-view data, similar to current diffusion based approaches (Ye et al., 2023; Shi et al., 2023a; Li
> et al., 2023; Shi et al., 2023b; Long et al., 2023)."
>
> This highlights the potential for future improvements by integrating pose-aware training techniques.
>
> **Q4.Question of the user study is ambiguous:**
>
> **A4:** Thank you for your question. We agree with your observation. In the revision, we revisited the user study experiments, evaluating semantic coherence and background consistency separately (see the screenshot in Fig. 18, Page 25). We have also updated Table 2 with the new results to reflect these changes.
> | Method                   | DDS   | Null Inversion | iRFDS (InstaFlow) | iRFDS (SD3) | iRFDS (Diffusion) |
> |--------------------------|-------|----------------|-------------------|-------------|-------------------|
> | **CLIP Score**           | 0.294 | 0.298          | **0.306**         | 0.297       | 0.296             |
> | **User Preference (Semantic)**    | 7.3%  | 20.1%         | **43.6%**        | 16.8%      | 12.2%             |
> | **User Preference (Consistency)** | 10.7% | 32.5%         | **33.4%**        | 13.1%      | 10.3%             |
>
>
> **Q5. Results of generated 3D assets (Figure 4) should be rendered in multiple views to be evaluated. Supp. materials have viewpoint-varying videos. Mentioning a Supp. would suffice.**
>
> **A5:** Thank you for your suggestion. We have included a link to the Appendix in Line 425.
>
> **Q6. Introduction is verbose. It would be clearer if non-essential sentences are removed.**
>
> **A6:**
> Thank you for your suggestion. In the revision, we have rewritten the introduction section to make it more concise, reducing its length by 10 lines.
>
> **Q7. Misc error**
>
> **A7:** Thank you. We have corrected it in the revision.
>
> **Q8. Why do the bottles change across n=1 and n=2 while boots stay the same in Figure 6?**
>
> **A8:**
> Thank you for your question. We think that optimizing 3D models introduces greater randomness because 3D models have more parameters than 2D images, and each time the rendered pose is randomly sampled, adding further variability to the process.
>
> **Q9. Why does the network Jacobian harm the results?**
>
> **A9:** Thank you for your question. At present, this remains a common empirical observation without a formal theoretical proof in liturature. We think one possible explanation is that the network learns a score function or flow velocity, which inherently encodes the gradient information.
>
> Additionally, we hope our response to Q1 offers further insights into this issue.

---

> > ### Comment · Reviewer_cYSo · 2024-11-24
> >
> > I thank the authors for their rebuttal. Most of my concerns are resolved.
> >
> > One last thing is the intuition behind finding multiple inverses from multiple noises, rather than using a single inverse. I am sorry that I wrote an ambiguous sentence in my initial review.

---

> > > ### Author Response · Authors · 2024-11-24
> > > **Thank you for your question.**
> > >
> > > Dear Reviewer,
> > >
> > > Thank you for your insightful question!
> > >
> > > This is indeed an option we carefully considered and explored for RFDS-Rev. We think the reasons are as follows:
> > >
> > > The inverse is conditioned on both the rendered image and the noise. To achieve single inverse, there are multiple choices.
> > >
> > > **1, The single noise choice: by using a single noise, we only need to inverse once for all poses.**
> > >
> > > **1.1** Defining a single noise for 3D is challenging. Since we use a 2D flow matching model, this requires assigning a specific noise to each pose of the 3D scene. However, this becomes difficult when accounting for the 3D geometry.
> > >
> > > **1.2** We also experimented with using a single 2D noise map for the entire 3D scene. However, in our case, we did not observe improvements by changing to this. Intuitively, consider a 3D point: if we sample a random camera pose, render the image, and apply the same fixed noise, the gradient for that 3D point remains random because it is on a random position on the rendered image. Additionally, we think even if a fixed 2D noise is used, each pose will inherently have different noise after the inversion process.
> > >
> > >
> > >  **2, The single image choice**
> > >
> > >  **2.1** A valid choice is to cache the noise after a single inverse for a specific pose and we do not need to fix the noise. However, in this case, we will be unable to randomly sample 3D poses, as there are infinitely many possible poses. There will be a space-time trade-off. We think this direction is worth exploring in future works.

---

> ### Author Response · Authors · 2024-11-26
> **Comment by Authors**
>
> Dear Reviewer,
>
> We hope our reply could address your questions.
>
> As the discussion phase is nearing its end, we would be grateful to hear your feedback and wondered if you might still have any other concerns we could address.
>
> We would greatly appreciate it if you could consider raising your rating of our paper.
>
> We thank you again for your effort in reviewing our paper!
>
> Authors

---

### Official Review · Reviewer_jHkr · 2024-11-04

**Soundness:** 4
**Presentation:** 3
**Contribution:** 3
**Rating:** 6
**Confidence:** 4

**Summary:**

The submitted work propose a way of utilizing rectified flow models a prior, that implies the internal knowledge of the model as in the form of an objective. Throughout the manuscript, the authors propose three different algorithms in this regard, where they are named as RFDS (Rectified Flow Distillation Sampling), iRFDS (inverse RFDS) and RFDS-Rev (RFDS Reversal). While discussing the analogy between the SDS (Score Distillation Sampling), which serves as a prior for models trained with score-matching objective (e.g. diffusion models), they demonstrate how these algorithms can be utilized as a loss objective reflecting the knowledge of the rectified flow model and the image inversion & editing task. Furthermore, the authors propose RFDS-Rev algorithm as the improved version of the baseline algorithm (RFDS).

To demonstrate the effectiveness of the set of the proposed algorithms, authors show the effectiveness of them on a variety of tasks. Initially, the authors demonstrate how does the proposed method improve over SDS in diffusion models in terms of generation quality. Following, they illustrate the applications of the proposed algorithms in inversion based tasks (e.g. image editing) and text-to-3D generation with a 2D prior from rectified flow models. For all of the demonstrated algorithms, authors provide qualitative and quantitative results that shows the effectiveness of the proposed framework.

**Strengths:**

- The paper proposed the first algorithm that utilizes rectified flow models as priors, to both enable implicit information encoded in the rectified flow model and inversion based image editing with such models.
- In addition to the baseline method provided, authors also propose an extension named RFDS-Rev, that improves over the baseline objective RFDS, that combines the algorithms proposed together and promises improved generation quality.
- Proposed method showcases satisfactory results on Text-to-3D generation, which shows the effectiveness of the proposed method over providing a prior on multiple views.
- Authors introduce a simplifying assumption enabling to efficiently using rectified flow models as priors, by simplifying the generator Jacobian. This serves as a modification for improving the efficiency of the proposed method.
- The paper provides sufficient amount of experiments demonstrating the effect of the design decisions made on the algorithms, which also serve as insightful observations (number of steps, effect of the Jacobian).

**Weaknesses:**

- In the examples provided, there is a significant saturation effect on the provided results (see Fig. 5, row 2 and Fig. 6, examples from SD3). It is unclear if that effect is a result of the proposed method or a property of the rectified flow models.
- While the image editing results seem semantically correct, there seems to be significant changes in the provided images (See Fig. 5) compared to methods such as Null-text Inversion. Despite the fact that the authors provide a user study and CLIP score based comparisons, the faithful reconstruction property is not discussed and it is unclear that how successful the provided approach is.
- In generation results provided in Fig. 7 (fox example), the generation results seem degraded in terms of quality and provides artifacts. It also seems that RFDS in general produces these artifacts. If the alternative method RFDS-Rev recovers those, authors need to provide related discussions and examples on its performance on the performed tasks (such as inversion based editing). If this algorithm is not applicable for such a scenario, it should be mentioned as a limitation of the method.

**Questions:**

- The authors are strongly encouraged to discuss more on the quality degradation issues mentioned in weaknesses. If this is an effect of the proposed algorithm, it should be discussed throughly.
- Authors should consider  extending the quantitative results on the reconstruction properties of the iRFDS. Despite the fact that the current evaluation assesses the edit in terms of success in reflecting the semantics, the qualitative results seems that the algorithm fails in preserving edit independent details.

---

> ### Author Response · Authors · 2024-11-21
> **Authors Response**
>
> Thank you for the insightful comments and constructive feedback!
>
> **Q1. Saturation effect on the provided results.**
>
> **A1.1: Saturation of iRFDS in Fig5.** Thank you for raising this important question. In our original submission, we employed the simplest implementation of the iRFDS loss without incorporating any specific tricks.
> This issue can be significantly mitigated by applying a small but commonly used trick—injecting the learned noise into an intermediate generation step (Hertz et al., 2022).
> In the revision, we address this issue in Lines 455–462 and provide a visual comparison in Appendix Sec. K and Fig. 12 (Page 20).
> Our observations show that the saturation issue is largely resolved.
>
>
>
> **A1.2: Saturation of 2D generation in Fig6.** The generation of over-saturated or degenerated images is a known issue caused by the  vanilla SDS loss.
> As explained in Sec. 3.4, our RFDS baseline is mathematically equivalent to the SDS loss when considering the PF-ODE formulation of the diffusion SDE.
> Consequently, the RFDS baseline inherits similar shortcomings.
> However, our proposed RFDS-Rev method significantly mitigates these issues (Fig. 3, Fig. 4) by applying a reverse process, converting images back into noise (Fig. 2, Sec. 3.3).
>
>
> **Q2. While the image editing results seem semantically correct, there seems to be significant changes in the provided images. the faithful reconstruction property is not discussed**
>
> **A2**: Thank you for raising this important question. Firstly, as mentioned in our response A1.1, background consistency can be improved by applying the insertion trick (Fig. 12 ,Page 20).
>
> Secondly, in the revision, we conducted a reconstruction test as detailed in Lines 463–467. Our observations indicate that by incorporating the same small trick used in prompt-to-prompt, our proposed iRFDS achieves a higher PSNR (28.96) compared to null-inversion (28.81).
>
> Additionally, we provide a visual comparison in Fig. 13 (Page 20) to further illustrate these improvements.
>
>
> **Q3. In generation results provided in Fig. 7 (fox example), the generation results seem degraded in terms of quality and provides artifacts. It also seems that RFDS in general produces these artifacts.**
>
> **A3**:
> Your observation is absolutely correct. The underlying reason aligns with our explanation in A1.2.
>
> In the revision, we updated Fig. 7 to include additional results from RFDS-Rev. These results clearly demonstrate that the proposed RFDS-Rev significantly improves generation quality and effectively resolves the degeneration issue.
>
> **Q4. The authors are strongly encouraged to discuss more on the quality degradation issues mentioned in weaknesses.**
>
> **A4**: Thank you. Please refer to our response of A1.2.
>
> **Q5. The qualitative results seems that the algorithm fails in preserving edit independent details.**
>
> **A5**: Thank you. Please refer to our response of A2.

---

> > ### Comment · Reviewer_jHkr · 2024-11-27
> >
> > Thanks to the authors for their responses to my questions. All of my concerns are resolved. Acknowledging that the presented method improves over the baselines with the inverse algorithm (Fig. 5) for the faithful reconstruction, I believe there is still a gap to be closed for satisfactory editing (in terms of quality). However, since the presented method provides an increment over the existing approaches, as demonstrated by the experiments, I lean more towards acceptance and keep my score as 6.

---

> ### Author Response · Authors · 2024-11-26
> **Comment by Authors**
>
> Dear Reviewer,
>
> We hope our reply could address your questions.
>
> As the discussion phase is nearing its end, we would be grateful to hear your feedback and wondered if you might still have any other concerns we could address.
>
> We thank you again for your effort in reviewing our paper!
>
> Authors

---

### Meta-Review · Area_Chair_c2io · 2024-12-19

**Metareview:**

This work proposes to distill knowledge from the pretrained rectified flow models, which serve as a powerful prior and can be plugged and played into various downstream tasks. The contribution is recognized as fundamental and the experiments are extensive, as commonly appreciated by reviewers. It receives three borderline accept and one borderline reject, but generally being consistently positive especially after the rebuttal. Several concerns raised by reviewers include more clarifications about figures/introduction, weak baselines, analysis about speed, etc. Authors provided detailed feedback via rebuttal and discussions, and reviewers mostly acknowledge that concerns are addressed well. AC agrees that it tackles a long-standing problem with a more fundamental solution that could impact and benefit many following works in this direction. After checking all the reviews and discussions, a decision of acceptance is made and authors are encouraged to include all revisions and newly added results/explanations in the final version.

**Additional Comments On Reviewer Discussion:**

There is no critical major concern raised. Several other concerns raised by reviewers include more clarifications about figures/introduction, weak baselines, analysis about speed, etc. Authors provided detailed pinpoint feedback via rebuttal and discussions, and reviewers mostly acknowledge that concerns are addressed well. The reviewer who gave 5 (marginally below) explicitly mentioned that all concerns are addressed. AC carefully checked it and inferred that the raised concern is not super critical and the tone should be on the positive side, and thus made the acceptance decision.

---

### Decision · Program_Chairs · 2025-01-22

Accept (Poster)